# From News to Forecast: Integrating Event Analysis in LLM-Based Time Series Forecasting with Reflection

**Xinlei Wang[1], Maike Feng[2], Jing Qiu[1], Jinjin Gu[1,*], Junhua Zhao[2,3]***

[1]School of Electrical and Computer Engineering, The University of Sydney
[2]School of Science and Engineering, The Chinese University of Hong Kong, Shenzhen
[3]Shenzhen Institute of Artificial Intelligence and Robotics for Society
{xinlei.wang, jeremy.qiu, jinjin.gu}@sydney.edu.au, zhaojunhua@cuhk.edu.cn

## Abstract

This paper introduces a novel approach that leverages Large Language Models (LLMs) and Generative Agents to enhance time series forecasting by reasoning across both text and time series data. With language as a medium, our method adaptively integrates social events into forecasting models, aligning news content with time series fluctuations to provide richer insights. Specifically, we utilize LLM-based agents to iteratively filter out irrelevant news and employ human-like reasoning to evaluate predictions. This enables the model to analyze complex events, such as unexpected incidents and shifts in social behavior, and continuously refine the selection logic of news and the robustness of the agent's output. By integrating selected news events with time series data, we fine-tune a pre-trained LLM to predict sequences of digits in time series. The results demonstrate significant improvements in forecasting accuracy, suggesting a potential paradigm shift in time series forecasting through the effective utilization of unstructured news data.

## 1 Introduction

Time series forecasting [18, 21] serves as an essential foundation for decision-making across a wide range of economic, infrastructural, and social domains [2, 13, 14, 16, 56]. The purpose of analyzing time series data is to decode the evolving relationships within complex, dynamic real-world systems. Traditional forecasting methods, while effective at identifying patterns in historical data, perform well when time series distributions remain consistent over time. However, they have limitations in addressing sudden disruptions or anomalies caused by external random events, and do not systematically connect complex social events with fluctuations in time series data. Integrating insights from real-world events and their effects on social and economic behavior is crucial for improving the reliability and accuracy of time series forecasting.

News articles provide crucial insights into unexpected incidents, policy changes, technological developments, and public sentiment shifts—factors that numerical data alone may not capture. Integrating news into forecasting enriches its inputs with context that closely mirrors the complexities of human behavior and societal changes. On the one hand, news offers a real-time snapshot of events, enabling the model to adjust predictions based on updated information. On the other hand, qualitative data from news sources enables the model to account for non-linear and non-numeric influences. By combining both quantitative and qualitative insights, the model can improve forecast accuracy, especially in rapidly changing environments, making it more reflective of real-world dynamics.

In this work, we propose a unified approach that embeds news and supplementary information into time series data using textual prompts. By fine-tuning large language models (LLMs) [54, 55, 66],

---

*Junhua Zhao and Jinjin Gu are the corresponding authors.
†Code and data are available at `https://github.com/ameliawong1996/From_News_to_Forecast`.

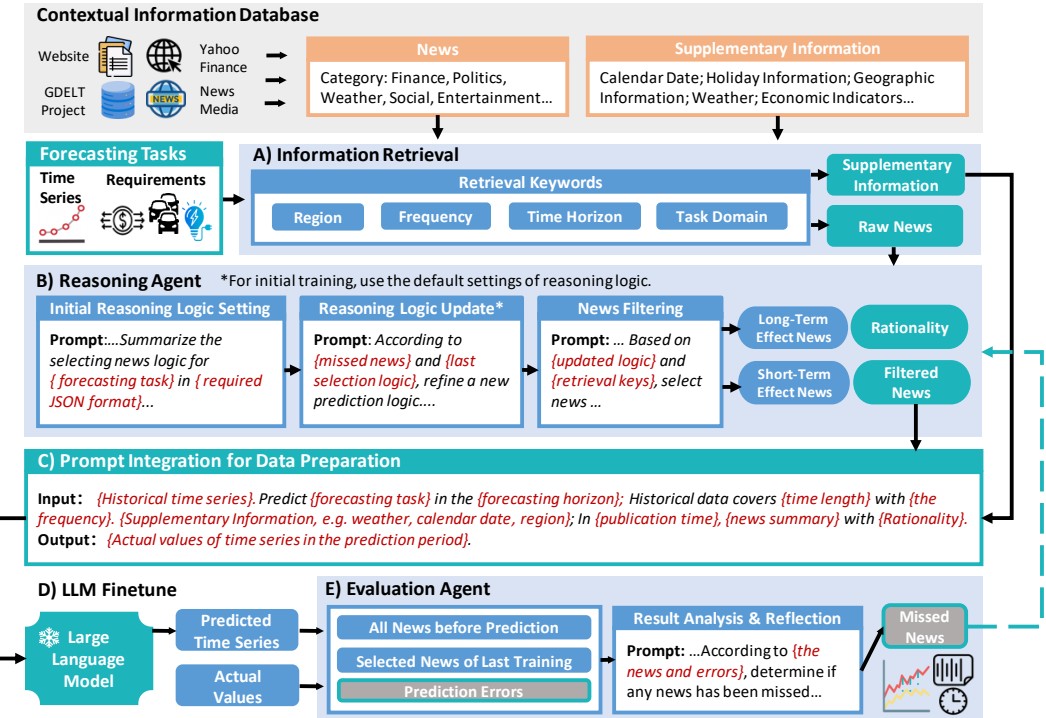

Figure 1: **Integrating textual information in time series forecasting**. (A) We retrieve relevant original news and supplementary information from our comprehensive database based on information such as the geographic location and time frame of the prediction task. (B) LLM-based agents analyze and select relevant news for different forecasting horizons. (C & D) The selected news and contextual information are combined with time series data for fine-tuning the LLM forecasting model. (E) Discrepancies between predictions and ground truth trigger a review of historical news and data to reprocess missed information and refine reasoning logic.

we transform time series forecasting into the prediction of the next token in text, taking into account relevant contextual information. The inductive reasoning capabilities of pre-trained LLMs, along with their ability to model multi-modal distributions, enable few-shot predictions in time series [17]. The potential of language models in time series forecasting has also been proven [23, 71]. After further training on our dataset, which includes time series, news, and supplementary information, language models can generate forecasts that consider the textual context provided in the input prompts.

Effective news filtering is a key issue for enhancing time series forecasting as input diversity increases. This task requires more than simple keyword extraction; it demands a deep understanding of how news elements interact with forecast variables, extending beyond linear reasoning to incorporate intelligent analytical methods. We employ LLM agents [43, 57] with advanced human-like reasoning capabilities for dynamic and effective news selection. These agents use few-shot learning to adapt their strategies based on logical scenarios that mimic human reasoning about factors affecting time series fluctuations. This enables them to identify relevant news, which is then paired with corresponding time series data to create a context-aware dataset that improves prediction accuracy. Additionally, LLM agents also play a crucial role in model evaluation, continuously refining their selection logic by comparing forecast errors with all relevant events. This iterative self-evaluation helps identify and integrate critical news items previously overlooked. By automating the analysis of unstructured text and applying chain-of-thought prompting, the agents effectively uncover patterns linking news events to forecasting discrepancies, revealing the nuanced effects of external factors on predictions.

Our contribution can be summarized as follows:

- The paper introduces a novel time series forecasting framework that integrates unstructured news data with numerical time series inputs, providing deeper contextual understanding and improving the model's responsiveness to social events and real-world dynamics.

- The research highlights the use of LLM agents for dynamic news selection and analysis. We leverage the reasoning ability of LLM to automate the effective understanding and filtering of

news content. The agents iteratively refine their news selection process based on forecasting results, improving model accuracy and reliability.

- We propose a data construction method that integrates time series data with news information, and build a dataset spanning multiple domains to support our research. The dataset includes task-specific time series data and verified public news reports, which facilitates further exploration in time series research.

- Our experiment results demonstrate the effective integration of news data, achieving superior prediction accuracy across diverse domains such as energy, exchange rate, traffic, and bitcoin domains. Our findings demonstrate that incorporating news is highly adept at navigating complexities, especially in energy demand patterns.

## 2 Related Work

**Time series forecasting.** The traditional method of time series forecasting relies on analyzing historical data and utilizing statistical models to predict future trends, with an assumption that past patterns will persist in the future [8, 12, 20, 27, 42]. However, these methods were limited to small-scale datasets. The advent of deep learning [30] has introduced a range of time series forecasting networks [33, 34, 36, 40, 53, 59, 60, 67, 69] that excel in managing larger, more complex datasets by capturing non-linearities and dependencies directly from historical data. Recent advancements include pre-training on diverse, large-scale datasets, allowing models to be fine-tuned on specific tasks with fewer data and resources [6, 24, 58, 63]. While these methods continue to evolve and improve performance benchmarks, they often neglect the influence of external and contextual factors.

Attempts to incorporate textual information (e.g., Twitter feeds, news articles, public reports) into time series forecasting have been made across various domains, such as finance [7, 48, 49], energy [3, 41], entertainment [26], pandemics [65], and tourism [47]. Traditional methods often simplify text analysis to counting keyword frequencies [41] or using dummy variables, which do not capture nuanced meanings. Advanced efforts include extracting richer textual features like word frequencies and sentiments using traditional NLP [9] and machine learning methods, as demonstrated by Bai *et al.* [3]. However, these approaches require labor-intensive feature engineering, struggle with long-text dependencies, and lack a deep contextual understanding. In contrast, LLMs excel in processing complex textual data and understanding contextual relationships, which can improve prediction accuracy and efficiency through automated feature extraction and enhanced scalability across multiple tasks. Despite their potential, no study has yet fully exploited LLM to enhance forecasting with their capabilities in understanding unstructured textual data.

**Language models for time series forecasting.** LLMs such as the GPT series [1, 4, 44, 45] and LLaMa [54] have excelled in a variety of natural language processing tasks. With their vast parameter sets, LLMs acquire extensive general knowledge and reasoning capabilities during pre-training, which is crucial for building intelligent systems equipped with common sense. LLMs architectures are increasingly applied to time series processing and forecasting [23, 25, 35, 37, 51]. For instance, TEMPO [6] adapts GPT architectures for dynamic temporal representation learning, while TIME-LLM [23] utilizes LLMs for time series forecasting by reprogramming input data and applying Prompt-as-Prefix techniques. Similarly, FPT [71] demonstrates that even frozen LLMs can perform effectively in time series tasks, leveraging the universality of self-attention mechanisms. Lag-LLaMa [46] uses a decoder-only transformer for univariate probabilistic forecasting, and Gruver *et al.* [17] show that by framing time series forecasting as next-token prediction, LLMs can surpass traditional models through effective tokenization and adaptation. However, existing studies mainly utilize the mapping capabilities of LLMs for numerical regression, without incorporating external textual inputs or leveraging the reasoning abilities of LLMs in understanding language.

**Reasoning with language models.** LLMs can automate tasks with human-like reasoning through "Chain of Thought" (CoT) prompting [57], enhancing reasoning by step-by-step emulation of human thinking [28, 32]. CoT prompting is useful for transforming complex questions into answers by introducing intermediate steps. The "Tree of Thoughts" (ToT) approach [61] refines this by mimicking trial-and-error methods, enhancing auto-regressive LLMs with a prompter agent, checker module, memory module, and ToT controller for multi-round dialogues. LLM-based agents can solve tasks by reflecting feedback signals in text and retaining them in a memory buffer for better decisions [50]. Cai et al. [5] proposed the LLMs As Tool Makers (LATM) framework, where LLMs create reusable tools for problem-solving, interleaving reasoning and actions to aid task completion and interaction [62]. These agents can debate their responses and reasoning to arrive at final actions [11].

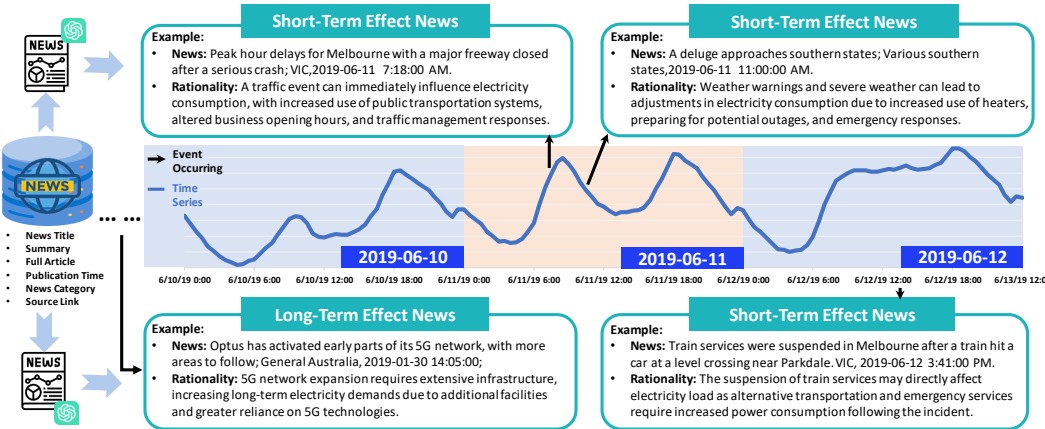

Figure 2: **Relationship between news and time series**. This figure illustrates the news filtered by the reasoning agents, using the example of Australia's state-level electricity demand. It features load data in Victoria state and selected news from June 10 to 12, 2019. The black arrow indicates time-specific events, the blue curve shows load fluctuations. The x-axis represents time, and the y-axis displays load values in kilowatts. The blue box displays the short-term impact news and long-term impact news selected by the reasoning agent (e.g., traffic incidents or new construction projects).

## 3 Method

In this work, we aim to integrate news insights into time series forecasting. The development of such a system faces several challenges. Firstly, the forecasting method must handle unstructured, non-numerical news inputs flexibly and adjust predictions based on the context of the news events. Secondly, constructing this model involves filtering news and establishing connections between the news and the time series data. This requires sifting through vast amounts of internet data to find relevant information, demanding deep societal understanding and sophisticated reasoning skills. Therefore, an intelligent agent is designed to manage this complexity. Moreover, potential inaccuracies in news selection or inferential errors may still affect the forecast accuracy, requiring further refinement of news selection and reasoning based on the predictions. Our approach includes three main modules: a language model-based forecasting module (Sec 3.1), a reasoning agent for news filtering and inference, and an evaluation agent to assess and refine the forecasting model (Sec 3.2). The core workflow of our method and the interrelations between these modules are shown in Figure 1. The subsequent sections will discuss these three modules in detail.

### 3.1 Rethinking Time Series Forecasting Problem and Elements.

**Time series forecasting** can be considered as a conditional generation problem of sequences [17]. This aligns with the general paradigm of natural language processing represented by LLMs. Taking the LLaMa language model as an example, assuming a number series {123,456}, LLaMa's tokenizer will regard this number as a sequence of digit tokens, *i.e.*, {"1","2","3"," ","4","5","6"}. Given the input series "123", the probability of predicting "456" can be represented as a probabilistic forecasting process in an autoregressive manner: $P(\text{"456"}|\text{"123"}) = P(\text{"4"}|\text{"123"}) \cdot P(\text{"5"}|\text{"4"}, \text{"123"}) \cdot P(\text{"6"}|\text{"45"}, \text{"123"})$. Generally, denoting the time series tokens at time $t$ as $x_t$, the LLM predict the next token in the series $x_{t+1}$ using the conditional probability distribution $P(x_{t+1}|x_{0:t})$. During pre-training, LLMs optimize its internal parameters to maximize this conditional probability over a wide range of natural language corpora. Though counterintuitive, Gruver et al. [17] have shown that pre-trained language models exhibit a significant few-shot capability for time series forecasting. This shows the potential of language models in understanding input digital tokens, and also inspires us to use the language model as a reasonable platform to study how to introduce the information contained in textual prompt into time series prediction.

**News context** offers critical insights into complex social events that traditional numerical data often overlook, and it also reflects sudden shifts in time series due to random events. In fact, the time series we collected can already be seen as being influenced by the aforementioned events. Assume an event $\mathcal{E}$ and a time series $x_{0:t}$, its impact on the future sequence can be expressed also as a conditional probability $P(x_{t+1}|x_{0:t}, \mathcal{E})$. However, when information about event $\mathcal{E}$ is not provided, we can only predict through historical time series. Although time series data itself can show patterns and trends, it

lacks the ability to indicate the causality behind events. The event information $\mathcal{E}$ offers the context needed to understand why certain spikes or drops occur. We show in Figure 2 some news that are closely related to time series forecasting.

In language models, news events can also be represented as text tokens. Consider a set of news text tokens $\{e_0, e_1, \ldots, e_u\}$, which represent event $\mathcal{E}$. LLMs treat this news information as a condition input and perform conditional probability predictions $P(x_{t+1}|x_{0:t}, e_{0:u})$. Including $e_{0:u}$ provides crucial contextual information that influences the prediction of future values. This process aligns with the standard approach used by language models to interpret text, allowing for the incorporation of information about multiple events through various news contexts simultaneously to enhance prediction accuracy. In practice, we only need to integrate news text with historical time series data in a prompt engineering manner.

**Other supplementary information** also provide contextual information for the forecasting model. For example, weather and climatic factors may affect energy demand and industrial output; financial indicators and economic metrics influence consumer behavior and business operations. Including this diverse range of information allows models to adjust for environmental, economic, and seasonal variations, improving prediction accuracy. The supplementary information can also be understood as conditions and integrated into the above conditional probability forecasting framework. Our approach incorporates this information into language for flexible integration with language models. For example, we use the text "*Weather on historical dates: the lowest temperature is 292.01; the highest temperature is 298.07; the humidity is 94.*" to express weather conditions. This enriches the input to cover various factors affecting the time series. Part (c) of Figure 1 illustrates the method of prompt integration for time series forecasting and the corresponding responses.

**Fine-tuning LLMs for time series forecasting.** Integrating the above information, we can construct inputs for LLMs to perform time series forecasting. Although pre-trained LLMs are already capable of generating time series predictions to some extent, relying on these pre-trained models for few-shot predictions in such a context-rich environment poses several challenges. Firstly, controlling the output of time series is difficult; predicting long sequences of numerical digit tokens is uncommon for LLMs. Secondly, the connections between the news and supplementary information and the time series typically need to be derived from historical data, which goes beyond the usual scope of using LLMs for few-shot time series predictions. To enable language models to more effectively forecast time series while considering the conditions imposed by news and supplementary information, we propose fine-tuning the language models to predict conditional probabilities. We employ a supervised instruction tuning method to train LLMs on historical time series data paired with corresponding news and supplementary information, formatted into text input-output pairs (shown in Appendix A.3). The same loss function used during pre-training is applied here. To fine-tune the LLM, we use the Low-Rank Adaptation (LoRA) method [19], which updates only a small subset of parameters, reducing computational demands while retaining most of the pre-trained knowledge. This strategy allows the model to efficiently adapt to new forecasting tasks without losing its foundational strengths.

## 3.2 Analytical Agent for Aggregation and Reasoning of Contextual News Information.

Next, we construct a dataset to train the above model. While obtaining time series data is relatively straightforward, matching it with appropriate news and supplementary information is not trivial. The internet is flooded with news, most of which are irrelevant to the time series we aim to forecast. Introducing irrelevant news can disrupt forecasting. Therefore, it is crucial to analyze the relevance and causality between the time series forecasting task and the select news accordingly. However, gaining such an understanding is complex, requiring knowledge of human societal mechanisms and logical reasoning skills. In our work, we utilize LLMs for filtering and reasoning about news content. We also recognize that even the most advanced language models cannot complete all reasoning and judgment in a single generation process. We explain how to use a combination of multiple LLM generations to create an intelligent agent that fulfills the complex requirements of news filtering.

**Time series and news pre-pairing.** For the initial stage of data preparation, news is retrieved to align with time series data based on matching time frequencies, horizons, and geographical areas. This synchronization ensures that insights from textual information are timely and regionally relevant. For example, to understand state-level electricity demand in Australia from 2019 to 2021, we gather local news from various Australian states and international news occurring in the same period that might directly or indirectly affect demand. In this way, potentially relevant candidate information can be roughly selected first, and such filtering can be easily completed through crawler means.

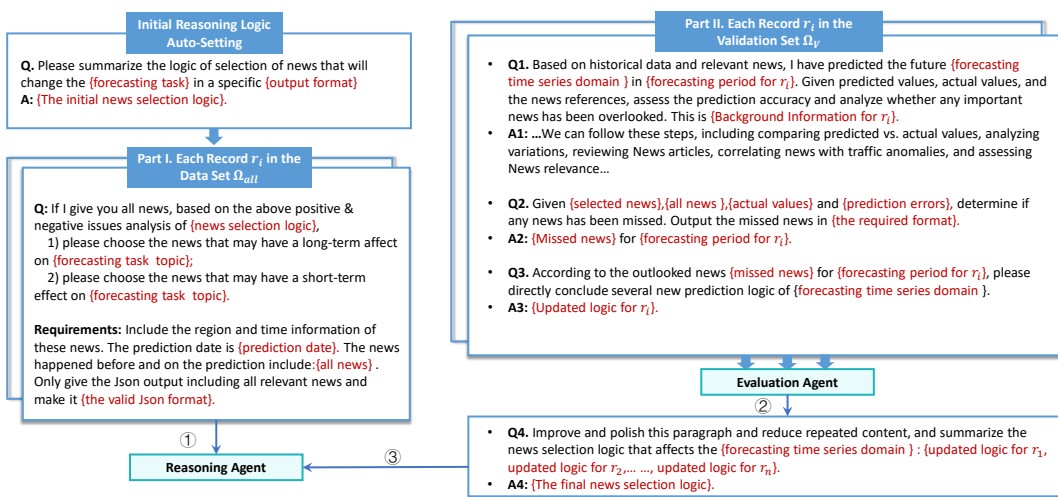

Figure 3: **Example of prompt designs for each iteration during fine-tuning.** Step 1 involves the reasoning agent selecting news using default logic. Step 2 evaluates predictions based on validation sets to refine the logic. In step 3, the updated logic directs data pairing for the next iteration. Full prompts are shown in Appendix A.6.

**Reasoning agent for news selection.** We employ an LLM-based reasoning agent capable of sophisticated tasks such as conversation, reasoning, and semi-autonomous action. This agent is programmed using detailed prompts that define roles, instructions, permissions, and context, enabling it to interpret human commands and perform complex tasks. This approach condenses extensive news datasets into a refined selection of pertinent articles. It leverages its reasoning capabilities to effectively screen, categorize, and interpret news texts. We employ few-shot prompting [4] and the CoT [57] method to develop an agent that understands the context of news for forecasting. This technique enhances the model's ability to handle multi-step, commonsense reasoning tasks essential for accurate forecasting. The agent uses multi-step prompts to break down complex problems into simpler, manageable parts. Our three-phase prompting method, illustrated in Figure 3 Part I, includes:

1. We develop an understanding of time series influencers, sorting them by impact (positive/negative) and duration (short/long-term), considering economic, policy, seasonal, and technological factors;

2. We direct the agent to filter and categorize news based on either automatically generated logic or a given reasoning logic, focusing on relevance to time series and classifying the impact (e.g., long-term and short-term) along with the rationale;

3. We specify the output format for the agent to organize the selected news into JSON, detailing aspects like summary, affected region, reporting time, and rationale. More details about our prompting method can be found in Appendix A.6.

The LLM agent can automatically develop an understanding of time series influencers, with the option to provide predefined reasoning logic in our models. In the automated process, the agent forms its logic through prompts designed to guide it in determining how different types of news impact a specific domain. For instance, we use open-ended questions to allow the agent to independently summarize and create its own filtering logic. User knowledge can also be incorporated into these prompts as additional reasoning, enabling the agent to generate more comprehensive logic. The agent then filters news based on the generated logic, whether fully automated or with user-provided input.

**Evaluation agent for reasoning updating.** We also design an evaluation agent to assess and improve the effectiveness of the aforementioned news filtering. Relying solely on the reasoning agent for news selection is not optimal, as the interaction between news and time series is complex. The reasoning agent can only analyze the potential impact of different news from the perspective of news content, without knowing whether the trained time series forecasting model based on them can make accurate forecasts. The evaluation agent is deployed after the time series prediction model has been trained. The evaluation agent extends beyond simple numerical assessments of prediction accuracy by incorporating human-like logical reasoning to refine the news selection logic chain. We focus our evaluation on identifying inaccuracies potentially caused by missing news, such as unusual events or illogical reports. It observes the model's predictive outcomes to determine if any crucial news has been overlooked and adjusts the news filtering strategy for the training data based on these results.

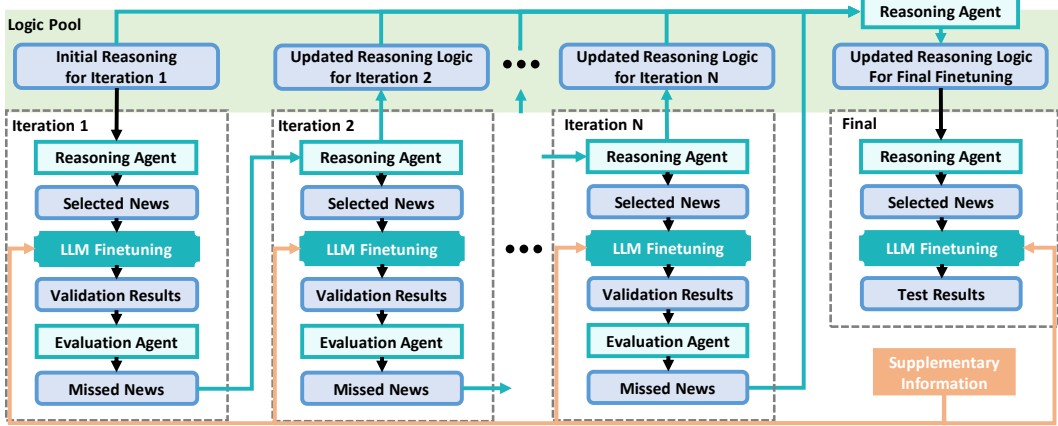

Figure 4: Overall pipeline iteratively combines news reasoning agents, fine-tuning, and evaluation agents.

For the evaluation agent, we structured the prompt design into three phases, as illustrated in Figure 3 Part II. In the first phase, we input the forecasting task type, the time horizon, and background information, which the agent uses to generate the steps for evaluating the prediction outcomes. In the second phase, we provide the ground truth, the discrepancies between predicted and actual series, as well as selected and historical news. The agent analyzes these inputs to identify overlooked news based on the distribution of prediction errors over time. In the third phase, the agent generates updated logic based on its analysis, guiding future news selection. After processing all validation set predictions, the reasoning agent consolidates the updated logic into a cohesive final strategy.

## 3.3  Overall Pipeline

We integrate the news reasoning and evaluation agents with the fine-tuning of the LLM forecasting model to enhance the quality of training data, as illustrated in Figure 4. In the first iteration, we use the LLM agent to establish news selection logic based on the domain and timing of the time series task. This logic directs the reasoning agent to filter relevant news, align it with time series data, and input it into the model for initial fine-tuning. After validating the model's predictions with a validation set, which is randomly extracted from the training data for each iteration, the evaluation agent checks for any missing news that may have influenced the prediction. This feedback helps the reasoning agent refine the filtering logic in subsequent iterations. The cycle continues until the final iteration, where the reasoning agent consolidates all updates to create the definitive news filter for training the final model. We use the GPT-4 Turbo model as the LLM for the agents described above.

# 4  Experiments

## 4.1  Data preparation

**Time series data.**    We selected time series data from domains influenced by human activities and social events to test our method's ability to capture complex human-driven dynamics during forecasting. These domains include Traffic [39] (traffic volume), Exchange [29] (exchange rate), Bitcoin [15] (Bitcoin price), and Electricity[15] (Australian electricity demand). To avoid bias from pre-trained language models, we updated the Exchange and Electricity datasets up to 2022. We use half-hourly electricity demand data from the Australian Energy Market Operator (AEMO) (`aemo.com.au`) and daily exchange rate data from the Exchange Rates API (`exchangeratesapi.io`). These datasets vary in frequency, including daily, hourly, and half-hourly updates, allowing us to evaluate the algorithms' effectiveness across different temporal resolutions. More details are in Appendix A.1.

**News collection.**    Since there are no public datasets that pair time series data with news events, we have collect news specifically for the above time series to facilitate our research. Some of the news content is collected from the GDELT dataset [31], a database tracking news from nearly every country in over 100 languages. GDELT provides real-time insights into societal, political, and economic events, enabling detailed analysis of global trends and their effects. We incorporate GDELT's event information into our forecasting models to enhance predictive accuracy. For domains needing the latest information, we collect real-time news from sources like News Corp Australia (`news.com.au`) and Yahoo Finance (`yahoo.com`), focusing on region-specific and task-specific activities.

**Supplementary information.** We enhance our forecasting models with open-source tools to grasp additional data for better accuracy and context. Weather information from OpenWeatherMap [10] provides daily temperatures, atmospheric pressure, wind speed, and humidity, crucial for load forecasting. Calendar dates, obtained using the Python packages `datetime` and `holidays`, account for seasonal and cyclical effects. Economic indicators are integrated using the `pandas_datareader` library, accessing data like GDP, inflation rates, and employment statistics from sources such as the Federal Reserve, World Bank, and international financial markets.

## 4.2 Results

**Effectiveness of news intergration.** In our approach, we incorporate news and supplementary information into time series forecasting by fine-tuning language models. Firstly, we assess whether this additional information can enhance time series forecasting. We conducted experiments to verify the necessity and effectiveness of integrating news data into our forecasting model. We compared four different scenarios as detailed in Appendix A.2 and Appendix A.3:

1. *Pure numerical tokens:* Uses numerical tokens, encompassing all variables without news. Except for region names or date information, it excludes other textual tokens as a baseline for comparison.
2. *Textual descriptive sentence tokens:* Evaluates whether using sentence-form descriptions instead of only raw digital numbers can enhance accuracy, with no news integration included.
3. *Unfiltered news with textual descriptive sentence tokens:* Assesses how integrating descriptive sentences of time series with unfiltered news data affects the model's performance.
4. *Filtered news with textual descriptive sentence tokens:* Shows the effects of integrating descriptive sentences with news that has been specifically filtered for relevance by the proposed agents.

Table 1: Performance comparison of different prompt designs. Red font indicates the best.

| | Electricity | | | | Exchange | | | |
|---|---|---|---|---|---|---|---|---|
| | RMSE | $\text{MSE}_{\times 10^{-3}}$ | MAE | MAPE | $\text{RMSE}_{\times 10^3}$ | $\text{MSE}_{\times 10^5}$ | $\text{MAE}_{\times 10^3}$ | MAPE |
| Only Numeric Prompt | 337.10 | 113.64 | 204.89 | 5.27% | 7.80 | 6.10 | 5.74 | 0.77% |
| Textual Prompt without News | 336.41 | 113.17 | 206.08 | 5.29% | 7.41 | 5.49 | 5.44 | 0.73% |
| Textual Prompt with Non-Filtered News | 407.86 | 166.35 | 250.75 | 6.84% | 8.28 | 6.86 | 6.37 | 0.85% |
| Textual Prompt with Filtered News | **280.39** | **78.62** | **180.96** | **5.15%** | **6.46** | **4.17** | **4.83** | **0.65%** |
| | Traffic | | | Bitcoin | | | |
| | $\text{RMSE}_{\times 10^2}$ | $\text{MSE}_{\times 10^3}$ | $\text{MAE}_{\times 10^2}$ | $\text{RMSE}_{\times 10^{-3}}$ | $\text{MSE}_{\times 10^{-6}}$ | $\text{MAE}_{\times 10^{-3}}$ | MAPE |
| Only Numeric Prompt | 4.55 | 2.07 | 1.66 | 4.46 | 19.94 | 3.07 | 5.72% |
| Textual Prompt without News | 4.44 | 1.97 | 1.54 | 3.87 | 14.97 | 2.76 | 5.08% |
| Textual Prompt with Non-Filtered News | 4.89 | 2.39 | 1.89 | 4.02 | 16.13 | 2.88 | 5.35% |
| Textual Prompt with Filtered News | **4.22** | **1.78** | **1.43** | **3.67** | **13.41** | **2.68** | **4.95%** |

The performance of different prompt designs is presented in Table 1. It can be seen that the fine-tuned LLM can be used to forecast time series even when using only digital tokens as prompts. The introduction of proper news and other supplementary information leads to significant performance improvements across all four domains. Nevertheless, such improvements are not easily achieved. If the introduced news information is not carefully selected, it can severely impair the results. There are two main reasons for this: first, the influx of a large number of news items introduces too many tokens, which can decrease the performance of the LLM as the number of tokens increases. Second, irrelevant news can introduce noise and incorrect causal information, leading to misleading predictions.

**Effectiveness of the evaluation agent.** To make our news filtering and reasoning processes more effective and comprehensive, we introduce an evaluation agent to reflect on and improve the effects of news selection based on prediction outcomes. As shown in Table 2, our evaluation agent refines news filtering through an iterative process, which is reflected in the progressively improved time series prediction results. The result corresponding to each case is the prediction outcome of the model after pairing the news selected according to the logic obtained from the corresponding iteration. Our findings suggest that, in most cases, two iterations are sufficient to achieve significant improvements, with multiple iterations consistently yielding better results than a single iteration due to the reflection mechanisms. We also found that this iterative filtering process reveals interesting insights into human society from the language model agent. These examples are presented in the Appendix A.7.

**Compare to other forecasting methods.** We also compare our method with existing time series forecasting techniques, detailed in Appendix A.1. We show a quantitative comparison against these baseline methods in Table 3. While the baseline methods use inverse normalization to revert predictions to their original scale, our model operates without normalization. This approach retains

Table 2: Comparison of Iterative Analysis. The baseline case is the initial selection. The arrow means the comparison of each case with baseline cases. A red downward arrow indicates an improvement, a blue upward arrow indicates performance degradation.

| | Electricity | | | | Exchange | | | |
|---|---|---|---|---|---|---|---|---|
| | RMSE | $MSE_{\times 10^{-3}}$ | MAE | MAPE | $RMSE_{\times 10^3}$ | $MSE_{\times 10^5}$ | $MAE_{\times 10^3}$ | MAPE |
| 1. Initial selection | 313.89 | 98.53 | 190.79 | 5.36% | 6.61 | 4.37 | 4.83 | 0.65% |
| 2. The second selection | ↓287.35 | ↓82.57 | ↓180.49 | ↓4.93% | ↓6.46 | ↓4.17 | ↓4.83 | ↓0.65% |
| 3. The third selection | ↓303.03 | ↓91.83 | ↑192.30 | ↑5.38% | ↑7.69 | ↑5.92 | ↑5.63 | ↑0.75% |
| 4. The fourth selection | ↓280.39 | ↓78.62 | ↓180.96 | ↓5.15% | ↓6.60 | ↓4.36 | ↓4.82 | ↓0.65% |

| | Traffic | | | Bitcoin | | | |
|---|---|---|---|---|---|---|---|
| | $RMSE_{\times 10^2}$ | $MSE_{\times 10^3}$ | $MAE_{\times 10^2}$ | $RMSE_{\times 10^{-3}}$ | $MSE_{\times 10^{-6}}$ | $MAE_{\times 10^{-3}}$ | MAPE |
| 1. Initial selection | 4.36 | 1.90 | 1.45 | 4.12 | 16.98 | 2.97 | 5.50% |
| 2. The second selection | 4.36 | 1.90 | ↑1.52 | ↓3.67 | ↓13.41 | ↓2.68 | ↓4.95% |
| 3. The third selection | ↓4.22 | ↓1.78 | ↓1.43 | ↓3.75 | ↓14.08 | ↓2.83 | ↓5.18% |

Table 3: Comparison of baselines for time series forecasting on different metrics. A lower value indicates better performance. Red: the best. Blue: the second best.

| Domains | Metrics | Ours | Auto. [59] | In. [67] | Dlin. [64] | iTrans. [38] | FiLM [68] | Times. [58] | Pyra. [36] | PatchTST[40] | FED. [69] | GPT4TS [70] |
|---|---|---|---|---|---|---|---|---|---|---|---|---|
| Electricity | MAE | 180.96 | 349.43 | 282.56 | 255.7 | 233.58 | 254.05 | 237.49 | 220.32 | 234.46 | 238.77 | 236.91 |
| | $MSE_{\times 10^{-3}}$ | 78.62 | 251.79 | 166.07 | 161.59 | 135.27 | 153.90 | 134.42 | 97.61 | 133.53 | 133.96 | 142.60 |
| | RMSE | 280.39 | 501.78 | 407.52 | 401.98 | 367.79 | 392.3 | 366.64 | 312.42 | 365.41 | 366 | 377.62 |
| | MAPE | 5.15% | 10.63% | 8.94% | 7.29% | 6.86% | 7.36% | 6.81% | 6.87% | 6.56% | 6.75% | 6.61% |
| Exchange | $MAE_{\times 10^3}$ | 4.83 | 9.27 | 1.75 | 6.96 | 5.12 | 6.44 | 5.24 | 14.6 | 6.73 | 8.98 | 15.05 |
| | $MSE_{\times 10^4}$ | 0.42 | 1.36 | 4.76 | 0.91 | 0.45 | 0.77 | 0.45 | 3.55 | 0.77 | 1.28 | 4.01 |
| | $RMSE_{\times 10^2}$ | 0.65 | 1.17 | 2.18 | 9.52 | 0.671 | 0.875 | 0.673 | 1.88 | 0.875 | 1.13 | 2.00 |
| | MAPE | 0.65% | 1.23% | 2.32% | 0.92% | 0.68% | 0.85% | 0.70% | 1.94% | 0.90% | 1.21% | 1.34% |
| Traffic | $MAE_{\times 10^2}$ | 1.43 | 2.49 | 4.44 | 1.70 | 1.56 | 1.65 | 1.61 | 1.51 | 1.84 | 1.74 | 1.64 |
| | $MSE_{\times 10^3}$ | 1.78 | 2.19 | 5.27 | 1.67 | 1.54 | 1.71 | 1.49 | 0.98 | 1.54 | 1.43 | 1.45 |
| | $RMSE_{\times 10^2}$ | 4.22 | 4.68 | 7.26 | 4.09 | 3.93 | 4.14 | 3.86 | 3.13 | 3.92 | 3.79 | 3.81 |
| Bitcoin | $MAE_{\times 10^{-3}}$ | 2.68 | 4.28 | 12.27 | 5.74 | 3.20 | 3.28 | 3.17 | 9.22 | 2.85 | 3.96 | 2.84 |
| | $MSE_{\times 10^{-6}}$ | 13.41 | 27.64 | 162.47 | 50.90 | 16.21 | 17.65 | 16.38 | 123.71 | 13.52 | 24.60 | 13.66 |
| | $RMSE_{\times 10^{-3}}$ | 3.67 | 5.26 | 12.75 | 7.13 | 4.03 | 4.20 | 4.05 | 11.12 | 3.68 | 4.96 | 3.70 |
| | MAPE | 4.95% | 7.61% | 21.28% | 10.39% | 5.70% | 5.84% | 5.64% | 16.16% | 5.13% | 6.97% | 5.08% |

the physical meaning of data, such as electricity demand or economic indicators, ensuring that our outputs remain interpretable. Normalization could obscure the impact of news events due to the nonlinear and scale-dependent relationship between these events and the original data values.

Our approach significantly outperforms traditional methods that rely solely on historical time series data in domains like electricity demand, exchange rates, and the bitcoin market, where events embodied in the news have substantial impacts. This demonstrates the potential of our method. However, the improvement from integrating news into the traffic sector is notably modest. The performance of our traffic forecasting model, which covers all roads in California, is hampered by the coarse granularity of publicly available news data, lacking the local details necessary for precise predictions. Traffic data primarily reflects specific road traffic flows, whereas our news sources are mostly regional or global, failing to capture localized traffic conditions adequately. This limitation is evident in the model's Mean Squared Error (MSE), which is sensitive to outliers and tends to exaggerate errors from traffic spikes that state-level news often does not report. Our model achieves good results with the Mean Absolute Error (MAE), indicating reliable average accuracy. Incorporating more localized road information could potentially improve these issues.

Figure 5 takes the electricity domain as an example and compares the ground truth with predictions by cases with or without news data, demonstrating the effect of incorporating news into forecasting models. The "With News" predictions are closer to the actual values than the "No News" predictions, particularly at critical timestamps where abrupt events significantly influence electricity demand.

## 5 Conclusion and Discussion

In conclusion, our study demonstrates the benefits of integrating news into time series forecasting using LLM-based forecasting method and LLM-based agents. These agents enhance model intelligence by autonomously identifying and addressing missed news, refining their logic, and assessing the impact of events on predictions. Our findings advocate for incorporating extensive domain knowledge, encouraging a shift towards more nuanced and context-aware forecasting. This approach enriches time series forecasting for adaptive, comprehensive forecasting aligned with real-world dynamics.

**Limitations of our approach.** While our approach demonstrates that LLMs like LLaMa 2 [55] can enhance time series forecasting by integrating news, there are limitations to its applicability.

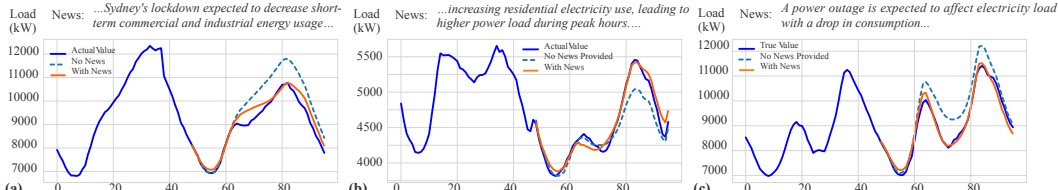

Figure 5: **Day-ahead Australia electricity demand forecasting with/without news.** The horizontal axis is the time index (half hour). Actual load demands are in solid blue, predictions with news in solid red, and predictions without news in dashed green. (a) Sydney's lockdown news effects; (b) Residential electricity consumption behavior news effects; (c) Anticipated power outage news effects.

The effectiveness of news integration is primarily evident in domains where human and market activities significantly influence trends. Our framework is less suitable for domains requiring precise meteorological modeling or where human activities have minimal impact, such as in meteorological or physical data. Additionally, the model is constrained by the maximum token length of pre-trained LLMs, complicating the simultaneous processing of large amounts of time series or multiple sequences, which can lead to data truncation and affect the accuracy of long-term predictions. Finally, our strategy enhances rather than completely replaces traditional time series tasks like classification or interpolation across all fields. Our aim is to demonstrate that by leveraging language models, it is possible to incorporate useful textual information to enhance time series prediction tasks.

**Future work.** Future enhancements will focus on extending the current forecasting model's scope in several key areas. Firstly, attribution analyses of news content used in the model will pinpoint which factors most significantly impact forecasting accuracy, facilitating an optimized news integration process. Advanced analytical toolkits can also be provided to reasoning agents, enabling sophisticated data processing and real-time application of complex analytical techniques. These developments will enhance the precision and relevance of the time series prediction model, contributing deeper contextual insights and expanding its applicability in the predictive analytics field.

**Broader impact.** Ethically, it is crucial that we conduct thorough reviews to ensure that our use of news content does not inadvertently perpetuate biases or negatively influence public opinion. This involves implementing rigorous checks for accuracy and balance to avoid the risks associated with misinformation, ensuring that our data sources are credible and that the content is factually correct. Furthermore, the potential misuse of news, especially the spread of "fake news", highlights the need for our models to incorporate sophisticated mechanisms to verify information reliability before integration. Beyond the discussed sectors, this approach has the capability to extend into forecasting GDP trends, analyzing carbon emissions, or predicting public health outcomes, each carrying significant implications for policymaking and public welfare. Thus, while our research offers substantial benefits in enhancing predictive analytics, it also obligates us to handle these capabilities responsibly, ensuring our contributions positively impact economic planning, environmental strategy, and informed decision-making across various domains.

# Acknowledgment

This work was supported in part by the Shenzhen Key Lab of Crowd Intelligence Empowered Low-Carbon Energy Network (No. ZDSYS20220606100601002); in part by the National Science Foundation Grant of China (72331009); in part by the Australian Research Council (ARC) Research Hub under Grant IH180100020; in part by the ARC Training Centre under Grant IC200100023; in part by the ARC Linkage Project under Grant LP200100056 and Grant ARC DP220103881.

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

# A  Appendix / Supplemental Material

## A.1  Experimental Setting / Details

Table 4: **Details of forecasting task designs and datasets.** The electricity data represents the half-hourly state-level electricity demand. The exchange rate data represents the daily exchange rate of the Australian dollar. Traffic data denotes the hourly traffic volume in California. Bitcoin data denotes the daily Bitcoin price.

| Datasets | Illness | Electicity | Exchange-Rate | Traffic | Bitcoin |
|---|---|---|---|---|---|
| Time Horizon | Not specified | 2019.01-2021.12 | 2018.01-2022.12 | 2015.01-2016.12 | 2019.01-2021.06 |
| Variates | 7 | 19 | 7 | 862 | 18 |
| Timesteps | 966 | 52,560 | 1,825 | 17,544 | 858 |
| Granularity | 1 week | 30 minutes | 1 day | 1 hour | 1 day |
| Input length | 24 | 48 | 7 | 24 | 7 |
| Prediction length | [24,36,48,60] | 48 | 7 | 24 | 7 |

**Baselines.**    To ensure a thorough evaluation of our model, we selected a diverse set of baselines that represent the latest advancements in time series forecasting, covering a broad spectrum of empirical studies. Our selection includes transformer-based models, such as transformer-based model: Informer [67], Autoformer [59], FEDformer [69], Pyraformer [36], PatchTST [40], iTransformer [38], FiLM [68]. Additionally, we included the CNN-based Timesnet [58] and MLP-based Dlinear [64]. Finally, we incorporated the LLM-based GPT4TS[70], leveraging GPT2's generative capabilities. For Informer [67], Autoformer [59], FEDformer [69], Pyraformer [36], PatchTST [40], iTransformer [38], FiLM [68], and Timesnet [58], we use the code from the `Time-Series-Library` (`https://github.com/thuml/Time-Series-Library/tree/main`). For the GPT4TS method, we use the official code from `https://github.com/DAMO-DI-ML/NeurIPS2023-One-Fits-All`. When conducting comparative experiments, all the information utilized in our method, except for the news content, is also provided to the baseline methods. While our LLM fine-tuning approach maintains the original scale of the numbers to preserve the physical meaning of the time series data, we normalize the variables to the range $[0, 1]$ for training the baseline methods. For non-numeric variables, we represent them using dummy variables for these baseline methods. To ensure optimal performance, we adhere to the official architecture settings for these methods and experiment with learning rates of 0.001, 0.0005, 0.0001, and 0.00005. Each configuration is tested three times with different initializations, and the best-performing setup is selected from all trials.

**Our method.**    When fine-tuning, we employ the LoRa [19] method to fine-tune the LLama 2 large language model [55]. The rank in LoRa is set to either 8 or 16, depending on the required token length to be processed. The alpha value in LoRa is set to 16, and the learning rate is set at 0.0001. When formatting numeric tokens, we uniformly maintain three significant figures. Retaining more significant figures offers minimal benefits and introduces an excessive number of tokens. The fine-tuning is conducted on a single NVIDIA A100 40G GPU, with the large language model undergoing several hundred to 1,000 training iterations on our curated data, requiring several hours.

**Evaluation.**    We choose five metrics to compare the prediction performances of different forecasting models. Mean squared error (MSE) is used to verify outliers, focusing on larger prediction errors. Root Mean Square Error (RMSE) measures error magnitude by averaging squared errors and taking the square root, aligning the error scale with the original data. Mean absolute error (MAE) is more balanced for different magnitude errors. Mean Absolute Percentage Error (MAPE) expresses the error as a percentage of the actual values, that is, on average how far off the predictions are from the actual values in percentage terms. The accuracy rate is defined as the proportion of true results among the total number of cases examined. These metrics are calculated by:

$$\text{MSE} = \frac{1}{n}\sum_{i=1}^{n}(y_i - \hat{y}_i)^2 \qquad \text{MAE} = \frac{1}{n}\sum_{i=1}^{n}|y_i - \hat{y}_i|$$
$$\text{RMSE} = \sqrt{\frac{1}{n}\sum_{i=1}^{n}(y_i - \hat{y}_i)^2} \qquad \text{MAPE} = \frac{100}{n}\sum_{i=1}^{n}\left|\frac{y_i - \hat{y}_i}{y_i}\right| \tag{1}$$

where $y_i$ represents the unnormalized true value of load, and $\hat{y}_i$ denotes the unnormalized result of the prediction model. $n$ denotes the number of prediction timestamps.

## A.2 Example of Numerical Input for Fine-tuning LLM

1. **Instruction:** "...7015.7,6875.1,6634.6,6334.6,6134.7,6007.9,..."

2. **Input:** "NSW; 2019-11-9; Weekend; not a public holiday; 2019-11-10; Weekend; not a public holiday;286.5;297.96; 34.0;1012.0; 284.92; 301.04; 46.0; 1016.0."

3. **Output:** "...6592.6,6467.0,6312.3,6066.8,5902.9,5795.0..."

## A.3 Example of Textual Input for Fine-tuning LLM

1. **Instruction:** "The historical load data is: ...7015.7,6875.1,6634.6,6334.6,6134.7,6007.9,..."

2. **Input:** "Based on the historical load data, please predict the load consumption in the next day. The region for prediction is NSW. The start date of historical data was on 2019-11-9 that is Weekend, and it is not a public holiday. The data frequency is 30 minutes per point. Historical data covers 1 day. The date of prediction is on 2019-11-10 that is Weekend, and it is not a public holiday. Weather of the start date: the minimum temperature is 286.5; the maximum temperature is 297.96; the humidity is 34.0; the pressure is 1012.0. Weather forecast of the prediction date: the minimum temperature is 284.92; the maximum temperature is 301.04; the humidity is 46.0; the pressure is 1016.0. On 2019-11-09 08:51:00, the news can change the time series fluctuation that The ongoing fires lead to an immediate and direct effect on today's load consumption mostly due to loss of infrastructure, increased demand from firefighting efforts, and the need for emergency communications. On 2019-11-09 20:20:00, the news can change the time series fluctuation that The devastating bushfires in NSW lead to increased short-term electricity consumption due to emergency services' operations, resident evacuations, and heightened communication needs. "

3. **Output:** "...6592.6,6467.0,6312.3,6066.8,5902.9,5795.0..."

## A.4 News Sources and Details

We analyzed the classical time series forecasting datasets, such as the Monash-TSF dataset, to identify the specific time periods covered by the time series data. For the traffic domain, data from 2015 to 2016 was used. For the Bitcoin domain, data from 2019 to 2021 was utilized. Relevant news information was filtered from the GDELT dataset using domain-specific keywords corresponding to these periods. We conducted web crawling and intelligent parsing of the associated news web pages. The traffic data was sourced from Yahoo, while the Bitcoin data was gathered from various websites. Regular expressions were employed to extract essential information from the parsed text, including news titles, URLs, publication dates, and the main content of the articles. From the GDELT dataset, we initially filtered 14,543 traffic-related articles and 19,392 Bitcoin-related articles. After removing invalid and redirected links, we retained 5,867 traffic articles and 5,906 Bitcoin articles. The traffic-related news data primarily covered the region of California, USA. The extracted data was formatted into JSON files to facilitate seamless integration into our model training.

For the region-specific tasks, such as AU electricity demand and AU exchange rate, we collected news articles spanning from 2015 to 2023 primarily sourced from news.com.au. In total, we gathered 380,560 articles covering a diverse range of topics pertinent to electricity demand and exchange rates in Australia. The news articles were processed using a similar methodology as described for the traffic and Bitcoin domains. We performed web crawling and intelligent parsing to extract key information from each article, including titles, summaries, categories, URLs, publication dates, and the full article contents.

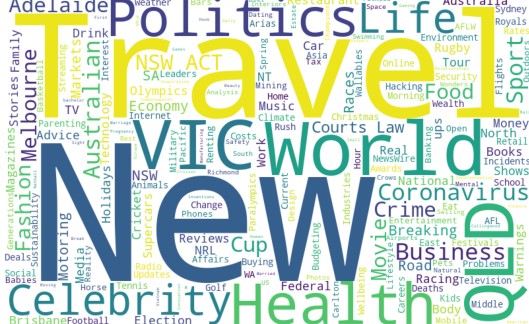

Figure 6: **Word-Cloud of News Categories.**

Regular expressions were utilized to ensure the accurate extraction of essential data points. The structured data was then formatted into JSON files, ensuring compatibility and ease of integration into our model training processes. By maintaining a consistent data collection and processing methodology across different domains, we ensured the reliability and utility of the gathered information.

Table 5: Statistical Information of Selected Event for Different Domains. In this table, we analyze the content and keywords of relevant news selected by the agent for training the model. We performed a word frequency analysis, with the second column showing the keywords and their frequencies in parentheses, and the third column representing the total number of selected news articles.

| Domains | Keywords of Selected News and Their Word Frequency | Total Number of Selected News |
|---|---|---|
| Electricity | emergency (339), weather (324), infrastructure (283), commercial events (181), residential (174), coronavirus (168), economic (160), heating (136), temperature (134), health (123), bushfires (115), construction (106), global (91), government (84), lockdown (76), traffic (71) | 3,655 |
| Exchange | economic (3033), international (2861), investor (1755), sentiment (1256), currency (1077), China (1058), trade (1049), USA (991), stability (886), geopolitical (786), financial (689), covid (652), health (440), political (328), energy (267), Trump (223), rba (186), tourism (172) | 4,383 |
| Traffic | California (1823), USA (1284), road (380), emergency (279), closures (275), commuting (233), police (233), disruptions (219), infrastructure (191), media (157), protests (149), congestion (138), incident (136), transportation (136), security (126), fire (124), wildfire (113), travel (91), shooting (85), vehicles (80), university (57), regulations (54) | 2,109 |
| Bitcoin | investor (1388), trading (557), sentiment (519), mining (449), global (262), interest rate (246), acceptance (240), Satoshi (212), security (209), economic (196), btc (177), nakaboto (150), institutional (140), Elon (132), technology (129), Ethereum (119), NVidia (114), geopolitical (108), bullish (100), legitimacy (97) | 2,616 |

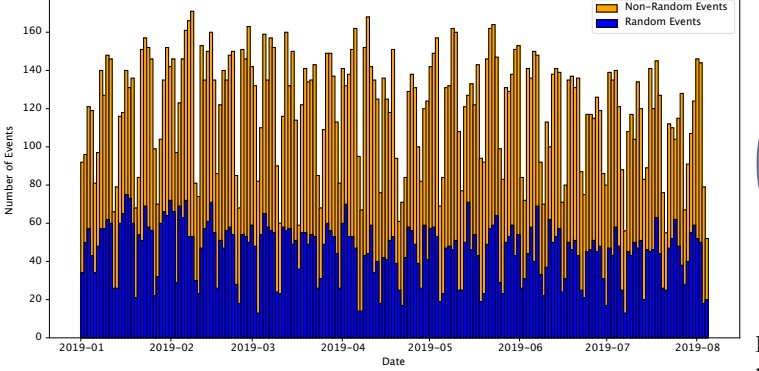

Figure 7: Daily Distribution of Random and Non-random Events.

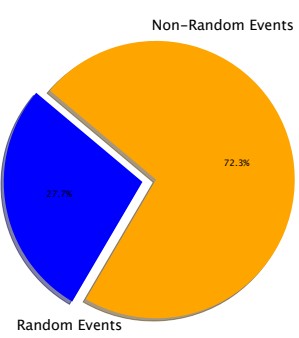

Figure 8: Pie Chart of the Proportions of Random and Non-random Events.

Random events are unpredictable and unplanned, such as natural disasters, accidents, health crises, and criminal acts. Normal events are planned or anticipated based on patterns, like political activities, sports & cultural events, economic reports, and public holidays. We used the LLM agent to categorize and detect all random and normal events from our raw news dataset spanning January 1st to August 5th, 2019. The analysis revealed that, on average, 27.7% of all events are random. Figure 7 and Figure 8 show the daily distribution of these random events. In our framework, the agent analyzes and selects the most relevant news, which mainly consists of five categories: economic or political events, health crises, natural disasters, technology development, and social sentiment. Additionally, the reflection agent, which analyzes prediction errors in the training dataset and missed news, helps identify unexpected and counterintuitive events buried in the raw news.

Table 6: We tested the effects of removing all or partial supplementary information, including Case 3, which includes only Filtered News, and Case 4, which includes Filtered News and Partial Supplementary Information. Removing all supplementary information and keeping only the news led to slight improvements compared to using only Supplementary Information in a numerical/textual prompt. However, this approach did not perform as well as Case 4. The results indicate that supplementary information remains essential. Overall, "Case 6: Textual Prompt with Filtered News and all Supplementary Information" which includes all supplementary information and filtered news, yields the best results.

| | Electricity | | | |
|---|---|---|---|---|
| | RMSE | $\text{MSE}_{\times 10^{-3}}$ | MAE | MAPE |
| Case 1: Only Numeric Prompt with all Supplementary Information | 337.10 | 113.64 | 204.89 | 5.27% |
| Case 2: Textual Prompt with all Supplementary Information (No News) | 336.41 | 113.17 | 206.08 | 5.29% |
| Case 3: Textual Prompt with Filtered News (No Supplementary Information) | 330.79 | 109.42 | 201.33 | 5.23% |
| Case 4: Textual Prompt with Filtered News and Partial Supplementary Information | 310.29 | 96.28 | 192.98 | 5.19% |
| Case 5: Textual Prompt with Non-Filtered News and all Supplementary Information | 407.86 | 166.35 | 250.75 | 6.84% |
| Case 6: Textual Prompt with Filtered News and all Supplementary Information | **280.39** | **78.62** | **180.96** | **5.15%** |

Table 7: We tried other language models using our proposed method. Mistral v0.1 [22], a 7B model, produced similar results as Llama 2 (7B). The Gemma 2B model [52] had slightly worse results, which may be due to its limited number of parameters. It is necessary to adjust the training for small models to achieve better results. Nonetheless, the results demonstrate the potential of language models to achieve good performance in our proposed methods.

| | Electricity | | | |
|---|---|---|---|---|
| | RMSE | $MSE_{\times 10^{-3}}$ | MAE | MAPE |
| Llama 2 (7B) | 280.39 | 78.62 | 180.96 | 5.15% |
| Mistral v0.1 (7B) | 291.66 | 85.07 | 206.08 | 5.35% |
| Gemma 2 (2B) | 350.08 | 122.56 | 216.72 | 5.63% |

## A.5 More Results

Figure 13, Figure 14, Figure 15, and Figure 16 provide showcases of some forecasting tasks results. It can be seen that our method has better results in predicting some sudden events and cases with distribution changes. We also tested the effectiveness of fine-tuned LLMs on traditional time series forecasting datasets. Given that long-term forecasting involves a significant number of digits and long sequences occupy many input tokens, we selected the Ill dataset for testing our method, which features prediction sequences of lengths 24, 36, 48, and 60. We also evaluated the performance of univariate forecasts and trained univariate versions of other baseline methods for comparison. The results, displayed in Table 9, show that, although counterintuitive, the fine-tuned LLMs can achieve performance comparable to baseline methods, providing a solid foundation for our research. Table 6 shows additional comparison experiments of different types of data incorporation, and Table 7 shows the comparison of using different pre-train language models.

## A.6 Full Prompt Design

### A.6.1 Prompt Example of Reselecting News through the Reasoning Agent:

**Prompt 1:** The reasoning logic is """Predicting Australia's region-level electricity demand every 30 minutes involves various factors:

1. Positive Issues Increasing Load:

   (a) Short-Term:

      i. Economic Growth: Boosts energy consumption.
      ii. Tech Advancements: New technologies spike demand.
      iii. Seasonal Factors: Extreme weather increases AC use.
      iv. Social Events: Large events boost energy use.

   (b) Long-Term:

      i. Population Growth: Raises residential energy use.
      ii. Industrial Development: Increases energy demand.
      iii. Urbanization: Expanding cities raise energy use.
      iv. Energy Transition: Shift to electric tech.

2. Negative Issues Decreasing Load:

   (a) Short-Term:

      i. Economic Downturns: Reduce industrial activity and energy use.
      ii. Efficiency Improvements: Lower consumption.
      iii. Weather Patterns: Mild weather reduces heating/cooling needs.
      iv. Public Health Crises: Reduce industrial and commercial activity.

   (b) Long-Term:

      i. Energy Efficiency: Better insulation and appliances.
      ii. Demographic Changes: Aging populations or lower birth rates.
      iii. Policy and Regulation: Promote conservation and sustainability.
      iv. Tech Innovations: More efficient technologies.

3. Other Factors:

(a) Political Stability: Affects energy policies and investments.

(b) Global Market Dynamics: Affect local energy prices and use.

(c) Environmental Consciousness: Changes in behavior and renewable adoption. """

**Prompt 2:** The prediction date is "2020-06-06". If I give you all news before the prediction, based on the above positive & negative effect analysis **<current_reasoning_logic>**:

1. please choose all news that may have a long-term affect on future load consumption

2. please choose all news that may have a short-term effect on future load consumption.

3. please choose all news that may have a real-time direct effect on today's load consumption. if there is no suitable news, please say no.

Also, please include the region (NSW/VIC/TSA/QLD/SA/WA) and the time information of this news. If there are multiple relevant news, please ensure that you include all relevant news. Remember to only give the JSON output, including all relevant news, and make it the valid JSON format.

**Prompt 3:** The news happened before the prediction include: **<all_news>**. The selected news is organized in JSON format. The Json output format is

```
{
  "Long-Term Effect on Future Electricity Demand": [
    {
      "news": "Another major renewable energy project was initiated in
          WA, expected to supply significant power by 2022.",
      "region": "WA",
      "time": "2019-03-15 11:30:00",
      "rationality": "Long-term electricity load will be impacted by
          the integration of renewable energy sources, which are
          expected to offset dependence on traditional fossil fuels."
    }
  ],
  "Short-Term Effect on Future Electricity Demand": [
    {
      "news": "SA just sweltered through a very warm night, after a
          day of extreme heat where some regional areas reached nearly
          48C.",
      "region": "SA",
      "time": "2019-01-03 17:57:00",
      "rationality": "Extreme weather conditions, particularly the
          intense heat, will lead to higher electricity consumption in
          the short term as residents and businesses increase the use
          of air conditioning and cooling systems to manage
          temperatures."
    },
    {
      "news": "A sudden cold snap in Victoria leads to a spike in
          electric heating usage.",
      "region": "VIC",
      "time": "2019-01-04 05:22:00",
      "rationality": "Short-term electricity load spikes are often
          caused by unexpected weather events that drive up heating or
          cooling demand."
    }
  ],
  "Real-Time Direct Effect on Today's Electricity Demand": [
    {
      "news": "An unseasonal downpour has wreaked havoc on Perth's
          electricity network this morning.",
      "region": "WA",
      "time": "2019-01-03 10:11:00",
      "rationality": "The sudden weather event causing disruptions to
          the electricity network can have an immediate impact on load
          consumption due to power outages, infrastructure damage, or
          emergency response measures."
```

```
    },
    {
      "news": "Lightning strike at a major substation causes
          widespread outages in Sydney.",
      "region": "NSW",
      "time": "2019-01-03 19:45:00",
      "rationality": "Direct effects on load consumption include
          sudden drops in power supply, triggering emergency measures
          to restore stability in the network."
    }
  ]
}
```

### A.6.2 Prompt Example of Evaluating Predictions through the Evaluation Agent:

**Prompt 1:** Based on historical data and relevant news from the last week, I have predicted the future exchange rate of Australia in the next day. The base is USD. I will provide you with our predicted values and actual values, as well as the news references. Please assess the accuracy of the predictions and analyze whether any important news has been overlooked.

**Prompt 2:** This is the background information: **<background>**

---

**Examples of Background Information**

" The start date of historical data was 2021-1-3, which is a Weekend, and it is not a public holiday. The data frequency is 1 hour per point. Historical data covers 7 days. The Daily GDP of Australia during the last week was 568773,568773,568773,568773,568773,568773,568773. The Daily Unemployment rate (unit: %) of Australia during the last week was 6.2556,6.2556,6.2556,6.2556,6.2556,6.2556,6.2556. The Daily Cash Rate Target (unit: %) of Australia during the last week was 0.1,0.1,0.1,0.1,0.1,0.1,0.1. The Daily GDP of the United States during the last week was 22600.2,22600.2,22600.2,22600.2,22600.2,22600.2,22600.2. The Daily Unemployment rate (unit: %) of the United States during the last week was 6.4,6.4,6.4,6.4,6.4,6.4,6.4. The Daily Interest rate (unit:%) of the United States during the last week was 0.1,0.1,0.1,0.1,0.1,0.1,0.1. The date of prediction starts on 2021-1-10, which is the Weekend, and it is not a public holiday. The Daily GDP of Australia on the prediction date is 568773. The Daily Unemployment rate (unit: %) of Australia on the prediction date is 6.2556. The Daily Cash Rate Target (unit: %) of Australia on the prediction date is 0.1. The Daily GDP of the United States on the prediction date is 22600.2. The Daily Unemployment rate (unit: %) of the United States on the prediction date is 6.4. The Daily Interest rate (unit: %) of the United States on the prediction date is: 0.1."

---

**Prompt 3:** This is the news we used for the prediction: **<selected_news>**; Here are all the news, including the day-ahead news and today's news in JSON format: **<all_news>**. Please determine whether there was any news that should have been considered in the prediction but was not included. The actual value is **<actual_values>**. Predicted values minus actual values are **<errors>**. The prediction date is **<predictions_time>**. According to the news and errors on this day, determine if any news has been missed. The output format should be: The missed news is xxx, occurred at xxxx, the possible reasoning is xxxx.

> **Examples of Missed News Outputs**
>
> **Answer:** Examining the provided news on 2021-01-09 for potentially unconsidered factors in the AUD/USD exchange rate prediction, one significant event stands out that could resonate with economic implications, specifically concerning Australia as a significant commodity exporter:
>
> - **Missed News Summary:** "Brisbane's streets have been empty in eerie scenes as the city experiences its first day of lockdown to stop the spread of coronavirus."
> - **Publication Time:** 2021-01-09 12:59:00
> - **Category:** Breaking News
> - **Possible Reasoning:** This news shows a potential shift in economic activity due to new COVID-19 lockdowns in a major Australian city, which could affect the AUD due to changes in both domestic consumption and possibly a short-term impact on business conditions. Lockdowns can create short-term economic disruptions influencing investor sentiment and risk perception towards the AUD, potentially missed in the prediction model.
>
> These items suggest an underestimation of contemporaneous socio-economic events that might indirectly influence the AUD/USD exchange rate through shifts in economic expectations and risk sentiment related to the management of COVID-19, which would have been highly relevant for predictions made for 2021-01-10.

**Prompt 4:** According to **<outlooked_news>**, please directly conclude several new prediction logic of the daily AUD exchange rate. (Remember not to output the detailed implementation plan)

> **Example Answers of News Reasoning Logic Updation**
>
> **Answer:**
>
> 1. **Increased Risk Aversion:** The lockdown in Brisbane may heighten risk aversion among investors, leading to a potential decline in the AUD as investors flock to safe-haven currencies like the USD. This can be anticipated as a prevalent reaction whenever there are renewed concerns about COVID-19 impacts.
>
> 2. **Domestic Consumption Impact:** With the Brisbane lockdown, domestic consumption could be considerably reduced. A decrease in consumer spending within Australia might result in lower economic output forecasts, potentially weakening the AUD.
>
> 3. **Market Sentiment and Speculation:** News of the lockdown and its impact on one of Australia's significant cities can drive market sentiments and speculative trades, which often have immediate effects on currency exchange rates. Negative news could prompt a bearish sentiment towards the AUD.
>
> 4. **Impact on Commodity Prices:** As a major commodity exporter, any perceived slowdown in Australia's economy due to lockdowns can affect global commodity prices, influencing the AUD. With a decrease in economic activities, demands for commodities might decline, causing a domino effect on the currency.
>
> 5. **Investor's Perception of Economic Management:** Prolonged or frequent lockdowns might affect international perceptions of Australia's management of economic challenges during the pandemic. This perception can influence foreign direct investment flows and currency strength.

### A.6.3 Prompt Example of Refining the Final Logic

**Prompt 1:** Improve and polish this paragraph to reduce repeated content and summarize the news selection logic that affects the Australian dollar exchange rate:**<all_updated_logic>**

**Prompt 2:** According to the given updated logic, please directly rephrase the current prediction logic and output the adjusted new logic. This is the current prediction logic that you need to adjust and improve:**<selection_news_logic_current>**

## A.7 Missed News Examples

By analyzing the relationship between prediction errors and news through the agent, we can identify missing news (such as events with indirect impacts) and adjust the agent's expectations of how news events affect time series. For example, the agent might discover that a certain type of incident has a relatively small impact on a specific region. These intelligent and flexible discoveries are detailed in the missed news summary. Here are some examples of missed news during the iterative analysis:

1. **Missed News: " A coalition of 62 countries is backing Australia's call for an inquiry into the origins of the new coronavirus. It comes as Trade Minister Simon Birmingham's request for a meeting with China went unanswered."** Occurred at: 2020-05-16 21:15:00. Possible reasoning: While the news about Australia pushing for an inquiry into COVID-19's origins and facing unresponsive behavior from China was considered, the geopolitical tension this implies may have broader implications. This nature of international relations potentially has consequences for the AUD, particularly in terms of trade volumes and investor sentiment, given China's role as a major trading partner. Given the prediction results, this suggests that any immediate effect of this news on the exchange rate by the prediction date (2020-05-17) was minimal or offset by other factors considered in the model.

2. **Missed News: "Saudi Arabia, the world's top oil exporter, aims to achieve net zero carbon emissions by 2060, as stated by Crown Prince Mohammed bin Salman."** Occurred at 2021-10-25 11:21:20. Possible Reasoning: Although this did not happen in Australia, a deeper analysis of its impact on the AUD could enhance the prediction framework. This shift towards sustainable energy from a major oil producer could influence global oil prices and economic stability. As a significant exporter of commodities, Australia's economy and the AUD could be affected. The zero deviation between predicted and actual values suggests either accurate forecasting by other models or a delayed effect on exchange rates. Further analysis may be needed to understand the long-term impacts on currency values.

3. **Missed news: " US Senator Bernie Sanders, one of the most vocal critics of Donald Trump, has launched his presidential bid to a crowd of thousands of supporters."** Occurred at 2019-03-03 04:30:00. The possible reasoning is that political announcements in a country like the US can significantly impact global financial markets, including forex. Bernie Sanders's candidacy news could have influenced market speculation and investor sentiment, affecting USD values and other major currencies. This detection suggests that incorporating global political news from major economies like the US might enhance the model's predictive capabilities. Significant US political events, like presidential candidacies, can influence global markets and forex currency fluctuations. Thus, overlooking major political news can leave a model vulnerable to missing market reaction surprises.

   (a) **Australian Housing Market News: Missed News: "Despite economic uncertainty throughout 2020, Australians confidently embraced the housing market pushing sales volumes beyond even 2019 levels."** Occurred at: "2021-01-17 10:00:00". Possible Reasoning: Although this news might seem domestically focused, the robust activity in the housing market can reflect economic confidence and potentially boost investor sentiment towards the AUD. Healthy real estate market metrics often attract foreign investment and can have favorable ripple effects on currency strength.

   (b) **Australian Government Seeking Pfizer Vaccine Information: Missed News: "The Australian government is seeking 'immediate' advice and information after Norway reported 29 deaths related to the Pfizer vaccine."** Occurred at: "2021-01-17 10:10:00". Possible Reasoning: News related to vaccine concerns could create short-term caution among investors, especially in contexts where the vaccine rollout impacts economic recovery prospects. If international observers view the vaccine issues as a slowdown to Australia's reopening or economic normalization, it could temper enthusiasm for the AUD.

## A.8 Selected News by Agents

The news is finally filtered out by the agent based on the prediction task, and the rationale is explained. As shown in Table8, each forecasting domain has a corresponding example of selected news.

Table 8: Examples of Selected News for Different Domains

| Publication Time | News Title | Rationale | Domain | Region |
|---|---|---|---|---|
| 2/20/2019 18:59:21 | Elon Musk Praises Bitcoin: 'Paper Money is Soon Going Away' - Ethereum World News | Publicly negative statements by national banks about Bitcoin can cause immediate but transient drops in its price due to sudden shock and reaction in investor sentiment. | Bitcoin | International |
| 3/16/2015 22:24:33 | Rookie Los Angeles police officer sought in fatal bar shooting | This real-time law enforcement incident could lead to immediate road closures, increased police activity, and media presence in the area, significantly impacting traffic flow and causing delays in the vicinity of the incident. | Traffic | LA, USA |
| 8/26/2020 16:49:00 | Hundreds of Victorian health workers have been stood down and told to isolate after an outbreak of COVID-19 at a Melbourne hospital. | This event does not directly affect today's load consumption significantly but implies a potential decrease due to the reduction in operational capacity and energy usage at the hospital. | Electricity | Melbourne, VIC |
| 3/19/2019 | Interest Rate Decisions: Decisions by the Reserve Bank of Australia to hold or raise interest rates can enhance the AUD's appeal by attracting global capital in search of higher yields, particularly in a global low-interest-rate environment. | Immediate reactions to interest rate decisions can cause significant short-term fluctuations in the AUD/USD exchange rate as global investors adjust their portfolios accordingly. | Exchange | AU |

Table 9: Comparisons of different forecasting methods on the Ill dataset.

| ILL | Metrics | Finetuning LLM | Autoformer | Crossformer | FiLM | MINC | Transformer | PatchTST | TimeNet |
|---|---|---|---|---|---|---|---|---|---|
| 24 | MAE | 0.6341 | 0.6593 | 1.4327 | 0.7150 | 0.7088 | 0.6495 | **0.5927** | **0.5946** |
|    | MSE | 0.6143 | 0.7239 | 2.9286 | 0.8164 | 1.0646 | 0.8201 | **0.5424** | **0.5626** |
| 36 | MAE | 0.6597 | 0.6301 | 1.2915 | 1.0599 | 0.6875 | **0.5610** | **0.6068** | 0.6602 |
|    | MSE | **0.6478** | 0.5917 | 2.3478 | 1.3918 | 0.6686 | **0.4971** | 0.6885 | 0.8109 |
| 48 | MAE | 0.7006 | 0.7246 | 1.2873 | 0.7331 | 0.9404 | **0.6186** | **0.6602** | 0.6996 |
|    | MSE | 0.7033 | 0.7376 | 2.2908 | 0.8099 | 1.0886 | **0.5744** | **0.6366** | 0.7237 |
| 60 | MAE | 0.7673 | 0.7948 | 1.3335 | 1.0400 | 1.1852 | 0.7102 | **0.6976** | **0.6656** |
|    | MSE | 0.8358 | 0.8811 | 2.4814 | 1.4858 | 1.6650 | 0.7017 | **0.6991** | **0.6835** |

## A.9 Reasoning Logic Examples

Figure 9, Figure 10, Figure 11, and Figure 12 show some Reasoning Logic Examples.

**Predicting each state's region-level load consumption data in Australia every 30 minutes involves understanding diverse factors:**

**Positive Issues Leading to Increased Load Consumption:**
  **Short-Term:**
  1.Extreme Weather: Heatwaves or cold snaps increase heating or cooling usage, surging electricity demand.
  2.Public Events and Holidays: Celebrations on days (like Australia Day or Vivid Sydney Festival, Christmas, or the sudden announcement of public holidays) spike electricity usage, which lead to increased domestic electricity usage as gatherings, lighting, and celebration preparations surge.
  3.Special Shopping Days: High retail activity on days like Black Friday and Boxing Day Sales causes increased electricity consumption.
  4.Emergencies and News Events: Significant events lead to higher usage of lighting and electronic devices.
  5.Power Outages and Public Gatherings: Blackouts and large social events like concerts cause sudden and significant surges in electricity load.
  6.Media-Covered Events and Sports: National emergencies, significant broadcasts, and sports events drive higher electricity consumption through increased public engagement and operational needs in transportation and hospitality sectors.
  7.Increased commercial and public space activity related to the easing of social distancing restrictions.
  **Long-Term:**
  1. Urban and Demographic Growth: Expanding populations and urbanization elevate base electricity demand.
  2. Electric Vehicles and High-Energy Technologies: Adoption of electric vehicles and technologies like data centers boost overall electricity needs.
  3. Construction and Infrastructure Projects: New developments require substantial energy, increasing long-term demand.
  4. Renewable Energy and High-Tech Industries: Transitioning to green technologies and integrating automation and robotics in industries also raise electricity consumption.

**Negative Issues Leading to Decreased Load Consumption:**
  **Short-Term:**
  1.Energy Conservation Initiatives: Power-saving programs and efficient grid management reduce peak demand.
  2.Event Cancellations and COVID-19 Restrictions: Major event cancellations and lockdowns lead to significant reductions in commercial and industrial power usage, partially offset by increased residential consumption.
  3.Behavioral Changes and Smart Grid Management: Public energy conservation efforts and advanced grid technologies smooth out demand curves and decrease overall consumption.
  **Long-Term:**
  1. Efficient Technologies and Renewable Resources: More efficient appliances and increased use of solar and wind energy reduce grid dependency.
  2. Sustainable Building Designs and Regulatory Frameworks: Net-zero building designs and enhanced energy policies decrease long-term energy needs.
  3. Economic Shifts: Changes from manufacturing to service sectors lower the overall energy intensity.
**Other Factors:**
  1. Socio-Economic and Regulatory Changes: Lifestyle changes, new policies, and incentives for renewable energy significantly influence energy consumption patterns.
  2. Technological and Societal Shifts: Evolving technologies and social norms continually reshape how and when energy is used, necessitating dynamic adjustments in energy forecasting.
  3. Special News Events: Major events such as Sydney's lightning strike, severe weather impacts, Outbreak of COVID-19, Start of Vivid Sydney Festival, flood events, and public responses to social issues have historically been overlooked in load forecasts, affecting their accuracy.

Figure 9: **News Selection Logic for AU Electricity Domain**

**Predicting the exchange rate movements for AUD/USD requires understanding a myriad of factors.**

**Positive Factors Leading to AUD Appreciation:**
    **Short-Term:**
1.Rising Commodity Prices: Australia's economy heavily relies on commodity exports; higher global prices can strengthen the AUD.
2.Interest Rate Decisions: Higher interest rates set by the Reserve Bank of Australia (RBA) can attract foreign capital, boosting the AUD.
3.Positive Trade Balances: An increase in export volumes relative to imports generally strengthens the local currency.
4.Investor Sentiment: Positive economic data or political stability can enhance investor confidence in AUD.

    **Long-Term:**
1.Economic Growth: Sustained periods of robust economic performance can enhance currency strength.
2.Foreign Direct Investment (FDI): Increased FDI into Australia bolsters economic capacity and currency value.
3.Global Positioning: Strengthening of Australia's strategic economic agreements with other countries.
4.Policy Stability: Consistent and favorable economic policies enhance investor confidence and currency value.

**Negative Factors Leading to AUD Depreciation:**
    **Short-Term:**
1.Falling Commodity Prices: Declines in global commodity prices can reduce the AUD's value.
2.Interest Rate Cuts: Lower interest rates by the RBA make AUD less attractive to foreign investors.
3.Negative Trade Balances: Higher imports than exports can weaken the AUD.
4.Market Volatility: Uncertain global financial conditions can lead to AUD depreciation.

    **Long-Term:**
1.Economic Slowdown: Long-term declines in economic performance can weaken the AUD.
2.Decreased FDI: Lower levels of foreign investment can negatively impact the economy and the currency.
3.Policy Instability: Erratic economic policies can reduce investor confidence in the AUD.
4.Geopolitical Issues: International disputes or regional instability can negatively affect the currency.

**Other Factors:**
1. U.S. Dollar Strength: As AUD/USD is a direct quote of the Australian dollar against the U.S. dollar, strengthening of the USD inherently weakens the AUD/USD rate.
2. Global Economic Trends: Shifts in global economic health, such as international recessions or booms, heavily influence AUD/USD.
3. Reserve Bank of Australia Policies: RBA's monetary policy and currency intervention practices significantly impact AUD valuation.

Figure 10: **News Selection Logic for Exchange Domain**

**To predict how news will affect Bitcoin's price, it's essential to consider the types of news and events that can cause fluctuations. Bitcoin's price can also be swayed by a range of short-term and long-term events, investor sentiment, and broader economic indicators. Here's a summary of key types of news and events that could affect Bitcoin's price:**

**Positive Influencers on Bitcoin Price:**
- Adoption by Companies and Countries: News about large companies or countries adopting or investing in Bitcoin can lead to price increases.
- Regulatory Approval: Positive regulatory news, for example, the approval of Bitcoin ETFs or favorable legislation around cryptocurrency, can increase investor confidence and drive up the price.
- Technological Advances: Improvements in blockchain technology, security enhancements, and innovations that increase the utility of Bitcoin can also contribute to price appreciation.

**Negative Influencers on Bitcoin Price:**
- Regulatory Crackdowns: Announcements of governmental crackdowns or negative regulations affecting Bitcoin or cryptocurrencies general can lead to price drops.
- Security Issues: News about hacks, security breaches, or thefts involving Bitcoin exchanges or wallets can decrease confidence and negatively impact price.
- Market Manipulation Accusations: Reports or rumors of price manipulation in the crypto markets can lead to significant price volatility and uncertainty.

**Other Factors Impacting Bitcoin:**
- Macro-Economic Indicators: Similar to traditional currencies, global economic trends can affect Bitcoin. Economic instabilities, inflation rates, monetary policy changes in major economies, and significant shifts in stock markets can influence Bitcoin's market.
- Public Sentiment and Media Coverage: General public sentiment driven by media coverage can greatly influence Bitcoin's price. Positive news can lead to price increases, while negative news can cause declines.
- Market Trends of Other Cryptocurrencies: Often, Bitcoin's performance is linked with that of other major cryptocurrencies. A surge or drop in other crypto coins can similarly affect Bitcoin's price.

**Predictive Considerations:** Predicting Bitcoin's price movements based on news requires not only understanding the news itself but also how these events fit into the broader economic, technological, and regulatory landscape. Analysts typically monitor a wide range of sources and use both quantitative and qualitative data to gauge market sentiment and potential price movements. Cultural perception and adoption levels in different regions also play significant roles.

Figure 11: **News Selection Logic for Bitcoin Domain**

**Adjusted Prediction Logic for Assessing Traffic Volume Changes in California:**

**1.  Weather Impact Analysis:**
-   Integrate real-time and anticipated weather conditions, particularly severe phenomena like wildfires, storms, and snow. Forecast immediate and potential effects on traffic, including road closures and decreased travel speeds due to hazardous conditions.

**2. Infrastructure Dynamics**:
-   Embed live updates and projected timelines concerning construction activities, road shutdowns, and significant infrastructure ventures. Utilize adaptive algorithms to dynamically forecast alternative route usage and resultant traffic flow variations.

**3. Event Driven Traffic:**
-   Harness data from local event schedules and public announcements to anticipate the traffic implications of major events such as sports games, concerts, or festivals. Predict substantial increases in local traffic volumes as participants commute to and from locations.

**4. Economic Indicators:**
-   Evaluate economic influences such as openings of new businesses, significant layoffs, or industry closures to determine shifts in commuting patterns, focusing on movements toward or away from economic centers.

**5. Transportation Policy Adjustments:**
-   Incorporate latest modifications in traffic rules or public transit changes into the models to predict changes in travel behaviors, timings, and route preferences.

**6. Emergency and Incident Response:**
-   Employ real-time monitoring to swiftly adjust traffic volume predictions in response to accidents, disasters, or other unforeseen emergencies that could result in road obstructions or necessitate evacuations.

**7. Political and Social Influences:**
-   Factor in the effects of political rallies, demonstrations, and changes in government policies on road occupancy and public transit, tailoring predictions to the anticipated influence and size of such events.

**Summary for Impact Prediction:**
-   Scope and Magnitude: Assess local versus statewide impact by determining the scale of occurrences to better predict scope
-   Timing Considerations: Analyze event timing relative to traffic peaks to anticipate their immediate impact on traffic.
-   Event Predictability: Differentiate between scheduled, routine events and unexpected emergencies to enhance predictive precision.
-   Geographic Considerations: Evaluate the geographic location of events to discern if impacts are localized or widespread, assisting in targeted traffic management.
-   **By refining these components, the predictive system is boosted in its capability to efficiently manage, respond to, and adapt to evolving traffic conditions across California, leading to improved traffic circulation and less congestion.**

Figure 12: **News Selection Logic for Traffic Domain**

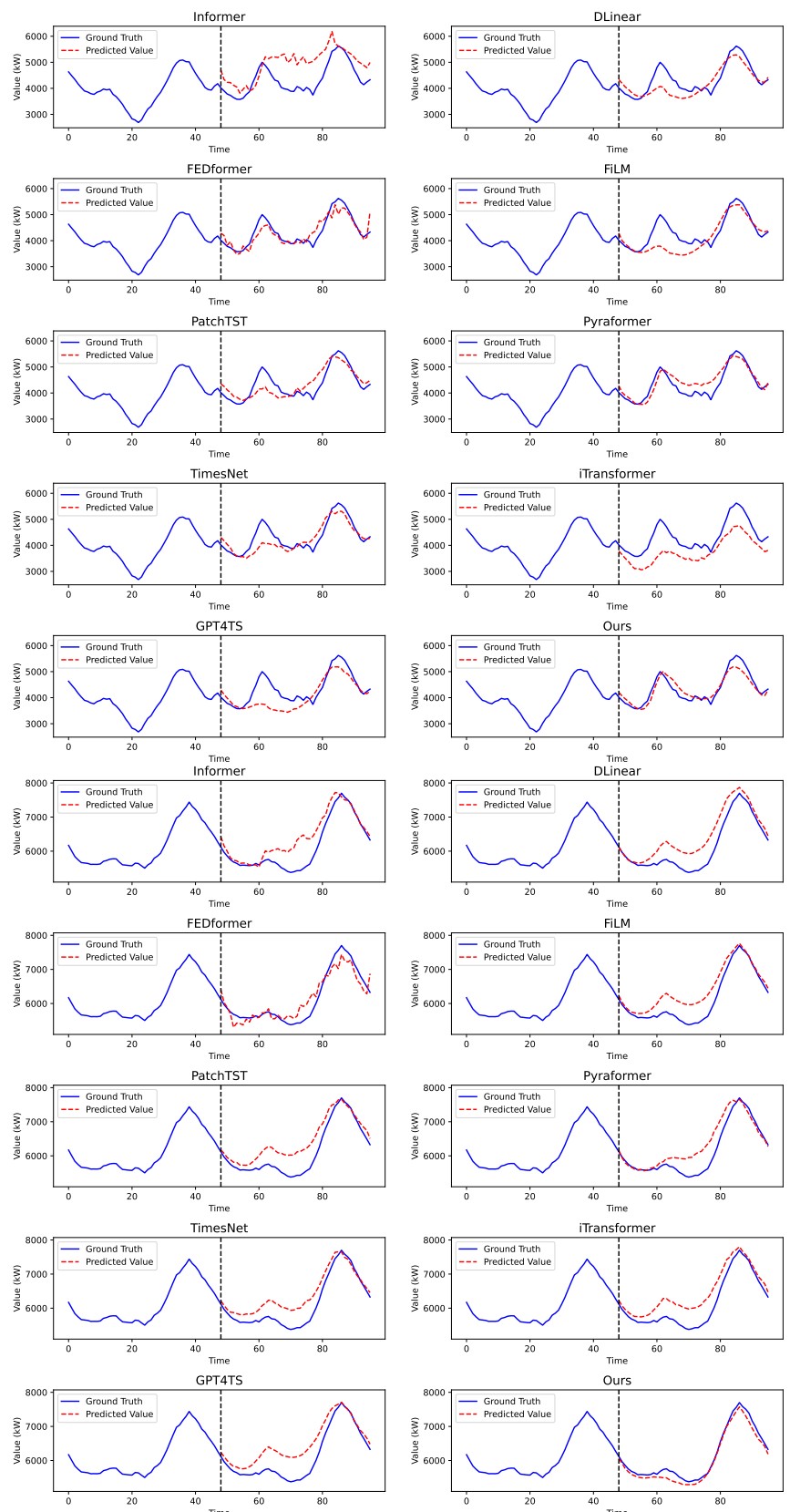

Figure 13: **Visualization of electricity demand forecasting results from different methods.**

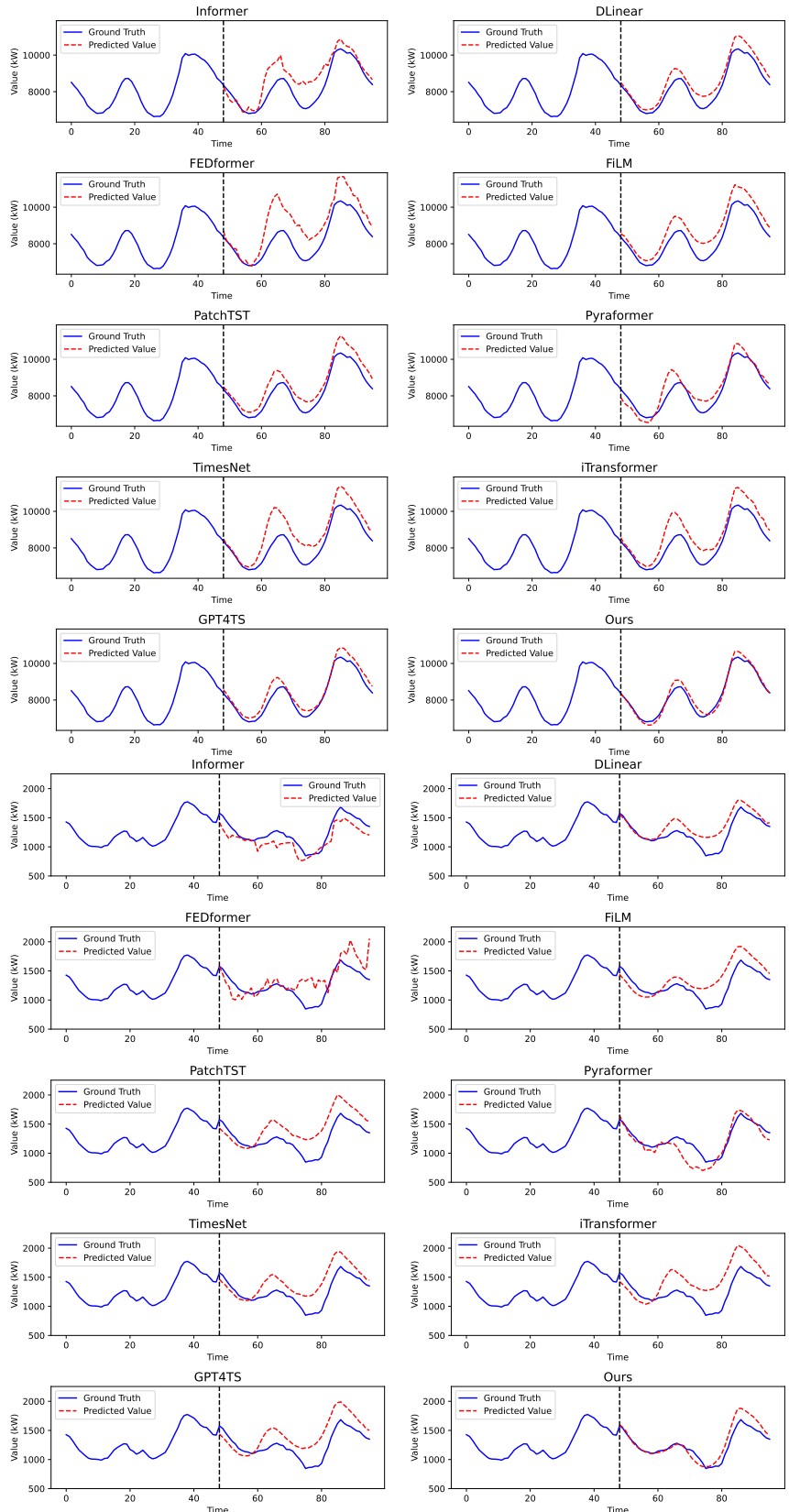

Figure 14: **Visualization of electricity demand forecasting results from different methods.**

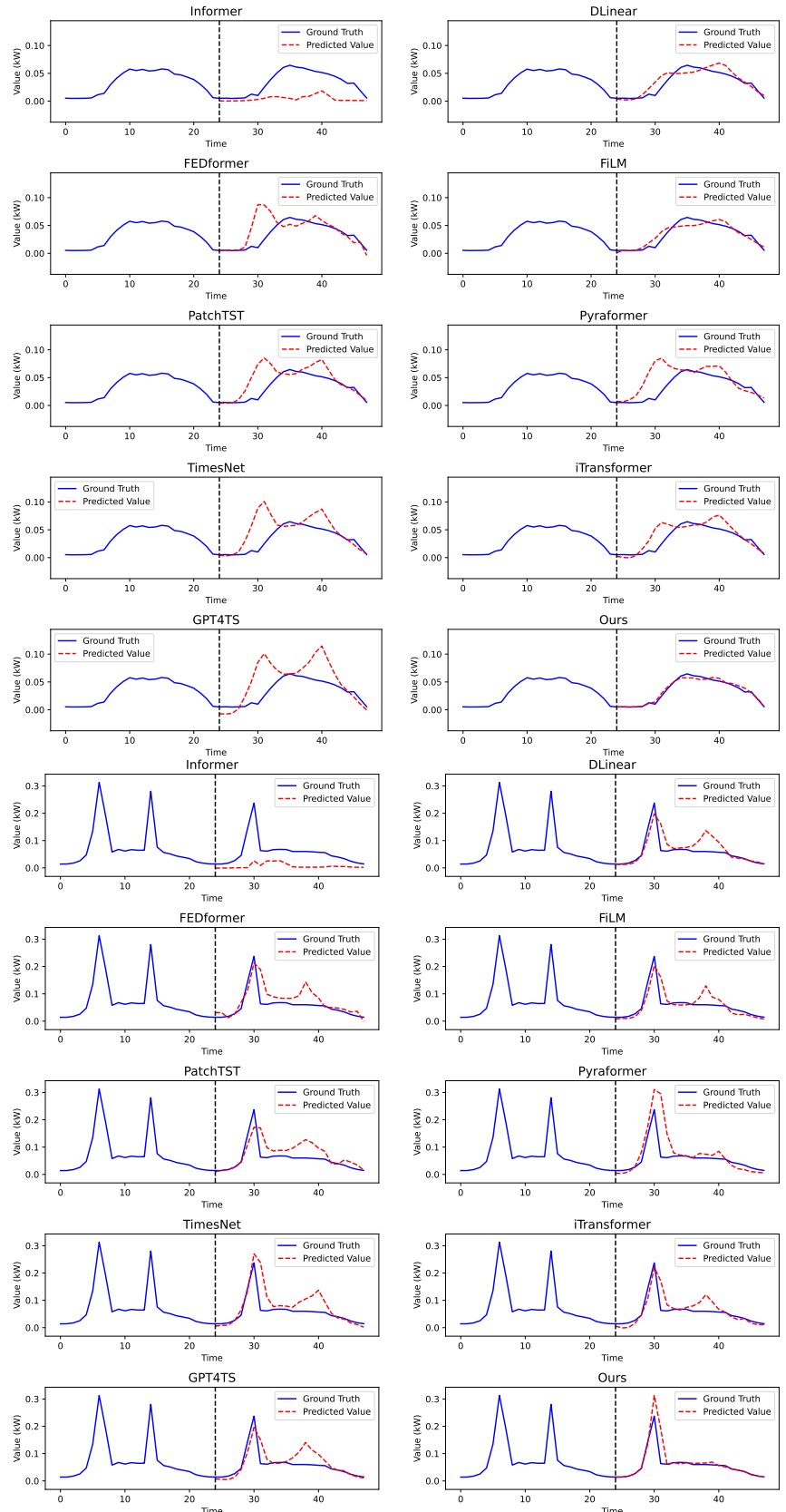

Figure 15: **Visualization of traffic volume forecasting results from different methods.**

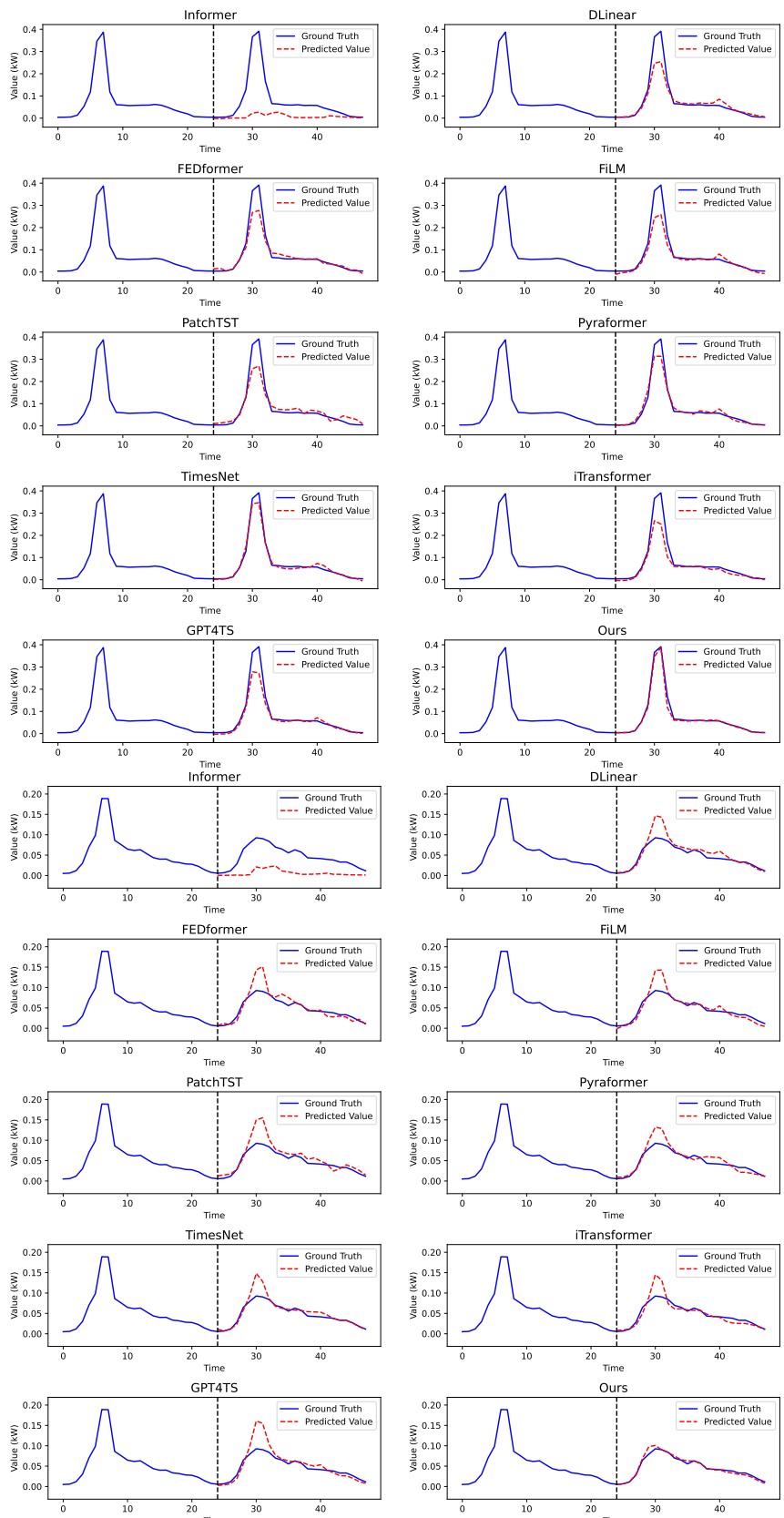

Figure 16: **Visualization of traffic volume forecasting results from different methods.**

