# OpenReview forum: "From News to Forecast: Integrating Event Analysis in LLM-Based Time Series Forecasting with Reflection"
_NeurIPS.cc/2024/Conference — NeurIPS 2024 poster_

### Official Review · Reviewer_pDM7 · 2024-07-11

**Soundness:** 3
**Presentation:** 3
**Contribution:** 3
**Rating:** 6
**Confidence:** 4

**Summary:**

This paper introduces a time series forecasting framework, where LLM-based agents are employed to sift out relevant news to time series of interests and the news are utilized to enhance the accuracy of time series forecasting models.

**Strengths:**

1. The idea of filtering and utilizing news to enhance time series forecasting is innovative and interesting.
2. The whole framework is well designed. Each component is with reasonable motivation.
3. The experimental result is impressive, showing great superiority of the proposed framework on time series forecasting.

**Weaknesses:**

1. Time consumption of the framework should be discussed.
2. The components in the framework are inherited from existing works, making the model itself not as innovative as the idea of the paper.
3. Collecting-then-filtering mechanism of the proposed framework might have negative effects on the real-time performance of the model.

**Questions:**

For real-world application, we may have to collect up-to-date news related to time series of our interests, do the authors have any suggestion of the amount of news?

**Limitations:**

Limitations are well discussed in the manuscript.

---

> ### Author Rebuttal · Authors · 2024-08-05
>
> We thank the reviewer for the insightful comments and for recognizing the innovation of our idea.
>
> ```Q1```: **''Time consumption''**
>
> ```A1```: Thanks for the question. The time cost of our method can be divided into training time and inference time. For a dataset with 3,500 time series, training requires approximately 10-15 A100 GPU hours, translating to 1-3 hours in a typical multi-GPU setup. Inference follows the normal language model’s reasoning time, with each instance taking from a few seconds to over ten seconds, depending on the output sequence length. With efficient information integration channels, obtaining real-time news can be very rapid, making real-time prediction feasible. We will include this information in the revised paper.
>
> ```Q2```: **''Model itself not as innovative as the idea of the paper''**
>
> ```A2```: Thanks for the question and for recognizing the innovation of our idea. We want to emphasize that our method is equally innovative. Our approach goes beyond model training to include **data construction methods, LLM agent construction methods, and a framework for integrating the agent with the model**. We innovatively combine these elements to utilize LLMs for time series forecasting, incorporating supplementary information and news. We introduced an LLM agent for reasoning and assistance within the LLM fine-tuning framework for time series prediction. We also proposed a Reflection mechanism to evaluate the LLM output of time series forecasting. These practices are novel in the literature and are crucial to our work. We believe these innovations will provide valuable insights and contribute to other projects leveraging LLMs to address domain-specific challenges.
>
> ```Q3```: **''Real-world application''**
>
> ```A3```: Thanks for your question. The amount of recent news required for practical applications varies based on the frequency, duration, domain, and geographical coverage of the specific time series forecasting tasks. Here are some general suggestions:
>
> **Amount of News**: For national-level events, we typically use an average of 100,000 news articles covering one year for model training. For week-ahead forecasting, about 1,000 articles covering one week provide a robust dataset, while day-ahead forecasting requires around 100 articles per day to cover a broad range of events. Our news sources include the GDELT [30], Yahoo Finance, and News AU.
> - Australia’s Half-Hourly Electricity Demand and Hourly Exchange Rate: We collected 380,560 raw articles/events over five years.
> - California’s Hourly Traffic Volume: We gathered 14,543 raw articles/events over two years.
> - Daily Bitcoin Price Prediction: We collected 19,392 raw articles/events from around the world over three years.
>
> **Real-Time News Collection**: We recommend using automated tools and APIs for real-time news collection. For example, GDELT 2.0 [30] updates every 15 minutes, capturing global news events in 65 languages, providing insights into various themes, emotions, and activities. This approach ensures an up-to-date dataset without manual intervention. It’s crucial to ensure the reliability of news sources by selecting official media channels and maintaining a sufficient number of high-quality, relevant, and diverse news items to improve the accuracy and reliability of time series forecasting.
>
> ```Q4```: **''Collecting-then-filtering mechanism might have negative effects on the real-time performance''**
>
> ```A4```: Thank you for your valuable feedback. The collecting-then-filtering mechanism is essential for ensuring the accuracy and reliability of the model's predictions. By initially collecting a broad range of news data to train the model, we use this mechanism to remove irrelevant news, thereby enhancing the overall quality of the input data for time series forecasting.
>
> While the training phase may require considerable time to analyze all raw news covering long periods in the datasets, the data collection process during the real-world testing phase can be automated, thanks to the availability of real-time news datasets, such as the GDELT Dataset [30] and Yahoo Finance API. On average, analyzing and filtering daily news requires 30 seconds to 1 minute, with each model's prediction instance taking from a few seconds to over ten seconds. This overall processing time is reasonable.
>
> In practical applications, there is often a trade-off between speed and accuracy. Although the collecting-then-filtering mechanism may introduce a slight delay, it significantly improves the model's accuracy and reliability. We believe that this trade-off is justified, especially in scenarios where precise and dependable predictions are critical. Our framework is also extensible and can be adjusted to improve real-time capabilities, and we will continuously explore further improvements to enhance both performance and accuracy.

---

> > ### Comment · Reviewer_pDM7 · 2024-08-12
> >
> > The response has addressed my concerns well and I decide to raise my score to 6.

---

### Official Review · Reviewer_UwQn · 2024-07-12

**Soundness:** 3
**Presentation:** 2
**Contribution:** 3
**Rating:** 6
**Confidence:** 4

**Summary:**

This paper proposes a new framework for time series forecasting. This framework fine-tunes a generative large language model (LLM) to improve forecasting accuracy by integrating news and supplementary information with numerical data and introducing iterative self-evaluation through LLM-based agents.

**Strengths:**

Strength 1: This paper has good originality to identify a unique challenge in time series forecasting task, which is the lack of effective modeling to address the distortions induced by additional random events with time going by.



Strength 2: Through Figures 1, 2, 3, and 4, this paper has good clarity to describe the proposed framework and the detailed procedures to complete the time series forecasting task.



Strength 3: This paper conducts comprehensive experiments, using time series datasets across multiple domains, to demonstrate the effectiveness of the proposed framework.

**Weaknesses:**

Weakness 1: There could be some statistical analysis on random events compared with normal events which represent a universal knowledge distribution with time going by.



Weakness 2: It seems a bit redundant and contradictory for methods 1) and 2). The description of the three-phase prompting could be better organized.



Weakness 3: Ablation studies and sensitivity analyses are encouraged. Based on the description of the four scenarios from line 290 to line 297, the news and the supplementary information are always integrated throughout the experiments.



Weakness 4: There could be more detailed analysis for Table 2, especially regarding the roles of the evaluation agent.

**Questions:**

Question 1: Apart from prediction accuracy, how to demonstrate the improved reliability? From line 144 to line 145, do sudden shifts embedded in random events from news context help the framework improve prediction reliability?



Question 2: From line 216 to line 217, is the understanding of time series influencers, or the sorting based on impact and duration, developed manually by people or automatedly by the LLM agent? What is the difference between such an understanding and a given reasoning logic mentioned from line 218 to line 219?



Question 3: From line 277 to line 278, apart from news articles, are there any ablation studies studying the accuracy differences between keeping and removing partial components of or the entire supplementary information?



Question 4: According to Table 2, why would introducing more rounds of reasoning selection decrease the forecasting accuracy?

**Limitations:**

The authors have adequately addressed the limitations and the potential negative societal impact of their work.

---

> ### Author Rebuttal · Authors · 2024-08-05
>
> Thanks to the reviewer for the insightful feedback and for highlighting the originality of our work.
>
> ```Q1```: **"Statistical analysis on random events compared with normal events"**
>
> ```A1```: Thanks for your question. To address it, let me first define random and normal events. Random events are unpredictable and unplanned, such as natural disasters, accidents, health crises, and criminal acts. Normal events are planned or anticipated based on patterns, like political activities, sports & cultural events, economic reports, and public holidays. We used the LLM agent to categorize and detect all random and normal events from our raw news dataset spanning January 1st to August 5th, 2019. The analysis revealed that, on average, 27.7% of all events are random. Figure 1 and Figure 2, presented in the rebuttal PDF, show the daily distribution of these random events.
>
> ```Q2```: **"The description of the three-phase prompting could be better organized"**
>
> ```A2```: Thank you for the feedback. We’ll clarify the distinction between Method 1, which covers the theory and mathematical modeling of how news affects time series predictions, and Method 2, which outlines practical steps for implementation, including dataset preparation, fine-tuning language models, and agent design. To improve clarity, we’ll present the three-phase prompting more clearly and concisely, moving from the conceptual idea to practical implementation, showing how these elements work together to enhance predictions. This reorganization will eliminate redundancy and improve the presentation.
>
> ```Q3```: **About ''prediction reliability''**
>
> ```A3```: Thank you for your insightful question. While sudden shifts in random events introduce volatility, our integration of agent reasoning helps the model effectively manage these changes. The agent does not simply translate individual events; it intelligently understands and processes news into concise summaries and rationales in standardized formats, focusing on the impact direction, impact duration, and impact scale of different events. This approach ensures consistent categorization of various random events as positive or negative, maintaining the model’s reliability. Additionally, we ensure prediction consistency through iterative reflections, demonstrating that the model produces stable results. Correlating prediction errors with missed news helps mitigate unexpected prediction deviations, further highlighting the model’s reliability.
>
> ```Q4```: **About line 216 -- line 219: "...difference between an understanding and a given reasoning logic..."**
>
> ```A4```: Thank you for your question. The LLM agent can automatically form the understanding of time series influencers, and providing a given reasoning logic in our models is optional. In the automated process, the agent forms its logic through prompts designed to help it determine how different types of news affect a specific domain. For example, we use open-ended questions to allow the agent to independently summarize and develop filtering logic. User knowledge can also be incorporated into these prompts as a given reasoning to help the agent generate more comprehensive logic. The agent then filters news based on the generated logic, either fully automatic or including user-provided input.  We will revise the paper to clarify these points.
>
> ```Q5```: **"Removing partial components of or the entire supplementary information"**
>
> ```A5```: Thanks for the question. We add more experiments with different training datasets (1. removing partial supplementary information; 2. removing the entire supplementary information). The results are shown in Table 2 in the rebuttal file.
>
> ```Q6```: **"Why more rounds decrease the accuracy?"**
>
> ```A6```: Thanks for raising this important question. Optimizing the number of iteration rounds is crucial for achieving the best prediction accuracy. Our findings in Table 2 of the paper show that multiple rounds generally enhance logic and provide more comprehensive insights compared to a single round, without inherently reducing accuracy. Typically, two or three rounds are sufficient enough for significant improvements. However, beyond this, additional rounds may introduce more complexity and noise, potentially affecting accuracy. In our study, we performed multiple iterations primarily to explore the potential for further enhancements and to understand the impact of iterative refinement. While we demonstrated the effectiveness of multiple rounds, determining the optimal number remains an ongoing challenge. This involves refining the evaluation agent’s workflow to balance reasoning depth with the risk of noise accumulation. Our future work will focus on finding this balance to ensure optimal predictive performance.
>
>
> ```Q7```: **"More detailed analysis regarding the evaluation agent"**
>
> ```A7```: Thanks for the question. The evaluation agent enhances news filtering by analyzing and relating prediction errors with potential missed news during training. It examines ground truth data, prediction errors, selected news, all raw news, and the forecasting task type to identify overlooked news. For example, if there is a significant discrepancy between the predictions and the actual values during a certain time window, it is necessary to closely examine the recent news from that time frame and look for any relevant events that may have been missed. This refines the model’s understanding of how missing news affects predictions. Insights from this analysis are then used to generate updated logic for subsequent news selection rounds, which is consolidated into a final version after processing all iterations for validation sets. Through this iterative evaluation, the model continuously improves its understanding of relevance. More details are in Section 3.2 and 3.3, Appendix A.6 and A.7.

---

> > ### Comment · Reviewer_UwQn · 2024-08-08
> >
> > Thank you for providing more detailed explanations on prediction reliability and conducting more experiments as ablation studies. I acknowledge that I have read the rebuttal and have no further questions.

---

### Official Review · Reviewer_MLnY · 2024-07-12

**Soundness:** 3
**Presentation:** 3
**Contribution:** 3
**Rating:** 6
**Confidence:** 3

**Summary:**

This paper introduces a novel approach to enhance time series forecasting using Large Language Models (LLMs) and Generative Agents. By integrating news content with time series data, the method aims to align social events with fluctuations in time series to provide enriched insights. The approach involves filtering irrelevant news and employing human-like reasoning to evaluate predictions, continuously refining the logic of news selection and the robustness of the model's output. The results show significant improvements in forecasting accuracy by effectively harnessing unstructured news data.

**Strengths:**

- The proposed framework integrates unstructured news data with numerical time series inputs, enhancing the contextual understanding and responsiveness to real-world events.
- The use of LLM-based agents for dynamic news selection and analysis is interesting. The agents effectively filter and analyze news content, continuously improving their logic based on forecasting results.
- The incorporation of news data significantly improves prediction accuracy across various domains such as finance, energy, traffic, and bitcoin, demonstrating the model's ability to navigate complex real-world dynamics.

**Weaknesses:**

- The performance heavily relies on the relevance of the news data selected. I'm worried that Inaccurate or irrelevant news can degrade the forecasting accuracy.
- The method may not perform as well in domains requiring highly localized or specific news data that is not available in general news sources.

**Questions:**

- How does the model ensure the relevance of the selected news items? Could the authors provide more details on the filtering criteria and logic used by the LLM agents?
- How does the model mitigate the impact of irrelevant or misleading news on the forecasting results? Are there any mechanisms in place to detect and exclude such news?

**Limitations:**

The authors have addressed some limitations, such as the dependency on the relevance and quality of news data and the complexity of integrating textual and numerical data.

---

> ### Author Rebuttal · Authors · 2024-08-05
>
> We thank the reviewer for the insightful comments and for recognizing the potential of our work.
>
> ```Q1```: **''Inaccurate or irrelevant news can degrade the accuracy.''**
>
> ```A1```: Thanks for the question. Inaccurate or irrelevant news does reduce prediction accuracy, as demonstrated in our paper. Reviewers can check Table 1 of our paper, where _the Case (Textual Prompt Non-Filtered News section)_ contains a lot of irrelevant news, significantly reducing the prediction effect and highlighting the importance of introducing the LLM Agent. Ensuring the relevance of news items is crucial for the effectiveness of our model. Our paper proposes LLM agent workflows (Section 3.2) to select and filter the most relevant news for predictions, thereby improving overall accuracy.
>
> ```Q2```: **"How does the model mitigate the impact of irrelevant or misleading news"**
>
> ```A2```: Thank you for your question. There are three ways to mitigate the impact of irrelevant or misleading news:
> 1. **News pre-process**: Before the agents analyze the news, we enhance relevance by ensuring source credibility and the temporal and spatial consistency between the selected news and the time series to be predicted. During the raw news collection process, we prioritize reliable and authoritative sources over less reliable ones. Additionally, relevant news is filtered from the open-source news dataset, considering spatial and temporal connections. For example, for traffic domain predictions, we primarily gather local news from California. We align news data with time series data by matching time frequencies, horizons, and geographical areas. This process selects the first round of raw news specific to each domain, ensuring the general relevance of the news to the time series being forecasted.
> 2. **LLM Agents**: We use LLM-based agents to filter and analyze news content, refining raw news into relevant events. Detailed methods are in Section 3.2, with specific prompts in Appendix A.6. Our agents autonomously filter and categorize news based on its impact (positive/negative) and duration (short-term/long-term). The selected news is formatted into structured JSON, detailing the affected area, time, and rationale. An evaluation agent continuously assesses and improves the filtering process by analyzing prediction errors, refining the model’s news selection, and excluding irrelevant or misleading news.
> 3. **News Verification**: Our extensible agent workflow can integrate fake news detection techniques to enhance reliability and accuracy. For instance, the agent can use fake news detection algorithms to assess the consistency and credibility of news content, effectively filtering out misleading information.
>
> ```Q3```: **"Filtering criteria and logic"**
>
> ```A3```: Thanks for your question. Our filtering logic for selecting news involves a multi-step reasoning process. First, our agent uses several Chain of Thought (CoT) prompts to understand the details of forecasting tasks and automatically identify time series influencers within the specific domain. This forms the initial filtering logic, which instructs the agent to sort news by impact (positive/negative) and duration (short/long-term), considering factors such as economic, policy, seasonal, and technological elements. The reasoning agent then filters and categorizes news based on this logic, focusing on its relevance to the time series and classifying the impact as long-term, short-term, or real-time. Additionally, the prediction time span is considered for further filtering of news. For instance, if day-ahead or real-time predictions are needed, long-term influencers will be filtered out.
>
> To refine this logic, we also deploy an evaluation agent that assesses prediction accuracy and uses logical reasoning to identify and correct inaccuracies from missing or irrelevant news. The evaluation agent works in three phases:
> 1. Input forecasting task type, time horizon, and background information to generate evaluation steps.
> 2. Analyze ground truth data, prediction errors, and selected news to identify overlooked news.
> 3. Update the logic based on its findings, refining it for future news selection.
>
> The refined filtering logic ensures the news data selected for time series forecasting is relevant and reliable, enhancing the model's predictive accuracy. For more details on the filtering logic examples and prompts, please refer to Section 3.2, Appendix A.6 and A.9.
>
>
> ```Q4```: **''Not perform as well in domains requiring highly localized or specific news data''**
>
> ```A4```: Thank you for your question. Relying solely on open-source datasets may not always be optimal for domains requiring highly localized or specific, unpublished news data. However, our paper demonstrates that including more relevant news can significantly enhance prediction accuracy, potentially improving traditional methods. While localized news may be hard to obtain, public events can still impact time series data related to human activities. Our results show that as the quality and relevance of news data increase, so does prediction accuracy. The proposed framework can also be adapted to incorporate localized and specific news sources, enhancing performance in specialized domains. Our paper demonstrates this potential of incorporating textual news into time series forecasting.

---

> > ### Comment · Reviewer_MLnY · 2024-08-13
> >
> > While I appreciate the thorough responses provided in your rebuttal, I have no further questions at this time and will maintain my positive rating.

---

### Official Review · Reviewer_QJWf · 2024-07-12

**Soundness:** 3
**Presentation:** 3
**Contribution:** 2
**Rating:** 6
**Confidence:** 4

**Summary:**

The paper proposes a novel method to integrate event news as external information into the time series forecasting system.

**Strengths:**

1. An important problem is studied in this paper.
2. An innovative idea of an automatic relevant news extraction mechanism is proposed.
3. Overall, the presentation is clear and good.

**Weaknesses:**

1. Some questions regarding the experiments need clarification.

**Questions:**

1. Though the idea that relevant event news may positively benefit time series forecasting is intuitively correct, this is not well demonstrated in the paper regarding the datasets used. Specifically, for many datasets tested, it’s hard to imagine what kind of news could dramatically affect them. Of course, it’s impractical to manually evaluate the news considered relevant. Besides the examples already provided, the authors could also test the average number of relevant news items per time window. This could give a rough idea of the distribution of relevant news, which can be used to approximate if the LLM works as expected.
2. Conflicting news could exist within a time window. For example, Elon Musk may praise or criticize a cryptocurrency within a short time frame. It seems that the model doesn’t consider this situation. Can the authors elaborate more on this issue?
3. The iterative analysis results show that the performance after each iteration is somewhat random. Though, in general, the final results are better than the initial iteration, it’s actually hard to predict if one iteration will be better or worse between two adjacent iterations. This raises a concern since the huge computational cost may seem unnecessary.
4. Another concern is that only LLaMA 2’s behavior is tested. Therefore, the sample set of LLMs tested is quite small, making it difficult to predict the behavior of other LLMs using the proposed method.

---

> ### Author Rebuttal · Authors · 2024-08-05
>
> We are grateful for the reviewer's insightful comments and for recognizing the novelty in our work.
>
> ```Q1```: **"Demonstrate the datasets used"**
>
> ```A1```: Thanks for your question. In Appendix A.4, we presented the source details of datasets, and they may answer part of the question. Our news datasets are collected from the GDELT Dataset [30], Yahoo Finance, and News AU. GDELT monitors a wide range of real-time worldwide web news. Specifically, (1) to predict electricity demand and exchange rates in Australia, we gathered 380,560 raw articles covering diverse national and international topics over five years; (2) to predict California's traffic volume, we obtained 14,543 raw articles covering two years in California; (3) For Bitcoin price predictions, we collected 19,392 worldwide articles covering three years. This collection of raw news data provides a foundation for analyzing the impact of various events.
>
> ```Q2```: **"What kind of news could dramatically affect"**
>
> ```A2```: In our framework, the agent analyzes and selects the most relevant news. According to our results, the selected news mainly consists of five categories: economic or political events, health crises, natural disasters, technology development, and social sentiment. For example, fiscal policy changes impact exchange rates, health crises like COVID-19 influence traffic volume and electricity load, AI breakthroughs can affect Bitcoin prices, and political events like elections or new legislation impact exchange rates and electricity demand.
>
> Additionally, the reflection agent, which analyzes prediction errors in the training dataset and missed news, helps identify unexpected and counterintuitive events buried in the raw news. These might include local incidents causing significant impacts or various events with indirect effects on time series. For example, news about Saudi Arabia’s net zero carbon emissions goal could impact global oil prices, indirectly affecting Australia’s economy and exchange rate. Although this news is not directly related to economic policy or typical keywords monitored, it indirectly reduces oil production for carbon mitigation, which can influence oil prices and exchange rates. Some analysis and examples can be found in Appendix A.7 and A.8. The reviewer can refer to these sections for more details.
>
> ```Q3```: **"Distribution of relevant news"**
>
> ```A3```: Thanks for the question. In the rebuttal file, we provide more statistical distribution details of selected news and their keywords in Table 1, Figure 1, and Figure 2. In our settings (mostly day-ahead prediction), the average number of relevant news influencing a specific domain generally includes 2 to 8 critical events per day. Several examples of selected news contents are provided in Appendix A.8.
>
> ```Q4```: **"Conflicting news could exist within a time window"**
>
> ```A4```: Thanks for raising this critical point. We respond to this question from the following aspects:
> 1. Our model does not simply understand individual events in isolation; instead, it considers the aggregation of various events within a specific timeframe. The model can synthesize conflicting information to better understand the overall trend. Different events cause immediate reactions in their respective timeframes. For example, different statements by Elon Musk on Bitcoin will have different effects on its price over time. Our model considers these factors (which are part of our input).
> 2. Our reflection agent identifies and adjusts conflicting news by analyzing prediction errors and all the raw news. The agent refines the model’s understanding of how conflicting news affects predictions, ensuring better adaptation to future events and improved accuracy.
> 3. When contradictory news appears, we may also need to judge its source and information authenticity. Our agent can flexibly integrate the analysis and reasoning process to determine the authenticity of new news. Reliable sources of news include official media and newspapers.
> 4. In the future, we plan to assign different weights to news based on source credibility, relevance, and context. For instance, statements from highly influential individuals will be weighted appropriately. If such conflicts appear, the reflection agent will adjust the weights accordingly.
>
> ```Q5```: **"It's hard to predict if one iteration will be better"**
>
> ```A5```: Thanks for the question. Our findings suggest that, generally, two iterations are sufficient to see significant improvements. Multiple iterations consistently provide better results than only one iteration due to the reflection mechanisms. In our study, we performed multiple iterations primarily to explore the potential for further enhancements and to understand the impact of iterative refinement. This process is crucial during the model training for refining the filtering logic. However, it is not required during the testing phase. For practical applications, the maximum number of iterations should be adjusted based on the prediction timeframe and the types of news being analyzed. For instance, if more categories of news or new trends emerge globally within the timeframe, additional iterations (three or four times) may be needed. Our paper demonstrates one possible approach for better prediction performance. In practical applications or engineering practices, the optimal iteration number can be determined by balancing performance gains against computational costs.
>
> ```Q6```: **"Difficult to predict the behavior of other LLMs using the proposed method"**
>
> ```A6```: Thanks for the question. We add more experiments with other LLMs, such as mistral 7b [R1] and gemma 2b [R2]. The results are shown in Table 3 of the rebuttal file, demonstrating the potential of these LLMs.
>
> [R1] Jiang, Albert Q., et al. "Mistral 7B." arXiv preprint arXiv:2310.06825 (2023).
>
> [R2] Team, Gemma, et al. "Gemma: Open models based on gemini research and technology." arXiv preprint arXiv:2403.08295 (2024).

---

### Author Rebuttal · Authors · 2024-08-06

We thank all reviewers and area chairs for their valuable comments. We are pleased that all reviewers have responded positively to our paper. They acknowledge that our work addresses an important problem (Reviewers QJWf, UwQn), introduces an innovative idea (Reviewers QJWf, UwQn), and demonstrates good results (Reviewer UwQn, pDM7). Additionally, they appreciate the clear presentation (Reviewer QJWf) and find the work both interesting and meaningful (Reviewers MLnY, UwQn, pDM7).

Reviewer QJWf (denoted as Reviewer 1), Reviewer MLnY (denoted as Reviewer 2), Reviewer UwQn (denoted as Reviewer 3), and Reviewer pDM7 (denoted as Reviewer 4) all give insightful comments. To answer their questions, we provide corresponding clarifications and analysis. Besides, we provide more experimental results. We summarize this rebuttal as follows:

1. According to Reviewer 1 and Reviewer 3’s questions, we add a statistical analysis of both selected news distribution and random event distribution during a specific time window. The results are illustrated in Table 1, Figure 1, and Figure 2 in the rebuttal PDF file.

2. According to Reviewer 1’s concerns, we add new experiments which use other language models with our proposed method. The results are in Table 3 of the rebuttal PDF file.

3. According to Reviewer 2 and Reviewer 3’s questions, we add more descriptions of our proposed method. We give a more detailed analysis of agent workflow for filtering irrelevant news.

4. According to Reviewer 3’s concerns, we add new experiments about the particular case of removing all supplementary information and partial supplementary information. The results are in Table 2 of the rebuttal PDF file.

5. Inspired by Reviewer 4’s comments, we discuss the time consumption and suggestions for the real-world application of our work in more detail.

In the final version, we will improve other minor points of Reviewer 1, Reviewer 2, Reviewer 3, and Reviewer 4. Thank you all for the valuable suggestions.

Best,

Authors

---

### Author Response · Authors · 2024-08-14
**Thanks to the ACs and the Reviewers.**

We sincerely thank the Area Chairs and the Reviewers for their time and effort during the discussion and review process. We are pleased to see that this paper has received consistently positive feedback from the reviewers.

In this work, we integrate news events into time series forecasting by fine-tuning a large language model. Additionally, we leverage LLM-based agents to iteratively filter out irrelevant news and apply human-like reasoning and reflection to evaluate predictions. We hope that this paper offers valuable insights into time series forecasting tasks and aids in the application of language models to real-world scenarios.

Bests,

The Authors

---

### Decision · Program_Chairs · 2024-09-25

**Decision:**

Accept (poster)

**Comment:**

This paper introduces a novel framework to integrate unstructured news with time series data for time series forecasting by using Large Language Models (LLMs). Empirical experiments demonstrate the effectiveness of the proposed framework by a noticeable improvement over baselines.

All the reviewers are positive about this submission. They also agree with the significant contribution of the paper such as novelty, strong empirical results. Concerns and issues raised by reviewers were properly addressed during the discussions with authors. They also promise to incorporate this improvement in the camera-ready version which we’re looking forward to.

Overall, this paper presents a solid and interesting work, I would love to recommend its acceptance .